



# High frequency of new particle formation events driven by summer monsoon in the central Tibetan Plateau, China

Lizi Tang[1], Min Hu[*,1,2], Dongjie Shang[1], Xin Fang[1], Jianjiong Mao[2], Wanyun Xu[3], Jiacheng Zhou[4], Weixiong Zhao[4], Yaru Wang[1], Chong Zhang[1], Yingjie Zhang[5], Jianlin Hu[2], Limin Zeng[1,2], Chunxiang Ye[1], Song Guo[1,2], Zhijun Wu[1,2]

[1]State Key Joint Laboratory of Environmental Simulation and Pollution Control, International Joint Laboratory for Regional Pollution Control, Ministry of Education (IJRC), College of Environmental Sciences and Engineering, Peking University, Beijing 100871, China

[2]Collaborative Innovation Center of Atmospheric Environment and Equipment Technology, Jiangsu Key Laboratory of Atmospheric Environment Monitoring and Pollution Control. Nanjing University of Information Science & Technology,

Nanjing 210044, China

[3]State Key Laboratory of Severe Weather & Key Laboratory for Atmospheric Chemistry of CMA, Institute of Atmospheric Composition, Chinese Academy of Meteorological Sciences, Beijing, 100081, China

[4]Laboratory of Atmospheric Physico-Chemistry, Anhui Institute of Optics and Fine Mechanisms, HFIPS, Chinese Academy of Science, Hefei 230031, Anhui, China

[5]School of Ecology and Nature Conservation, Beijing Forestry University, Beijing 100083, China

*Correspondence to: Min Hu (minhu@pku.edu.cn).*

## Abstract

New particle formation (NPF) is an important source of cloud condensation nuclei (CCN), which affects Earth's

radiative balance and global climate. The mechanism and CCN contribution of NPF at the high-altitude mountains, especially in the Tibetan Plateau (TP), was unclear due to lack of measurements. In this study, intensive measurements were conducted at Nam Co station (4379 m a.s.l) in the central TP during both pre-monsoon and summer monsoon seasons. The frequencies of NPF events exhibited extreme distinction with 15% in pre-monsoon season and 80% in monsoon season. The level of organic vapours governed the occurrence of NPF events, while condensation sink and gaseous sulfuric acid had no

effect. The frequent NPF events in summer monsoon season resulted from the higher concentration of organic vapours, which was brought from northeast India by the strong southerly monsoon. It had increased the aerosol number concentrations and CCN at supersaturation of 1.2% by more than 2 and 0.5 times compared with those in pre-monsoon season, respectively. Considering that the smaller particles formed by NPF may further grow and reach CCN size during the

following days due to the low-level coagulation sink, the amount of potential CCN in monsoon season could be much larger

than our local measurement results. Our results emphasized the importance of organics to NPF in high-altitude atmosphere,

and the seasonal effect of NPF at the high-altitude sites should be carefully considered in model simulations to reduce the

uncertainty of global CCN budget.

## 1 Introduction

Atmospheric aerosols can affect Earth's radiative balance through direct interaction with solar radiation and by serving

as cloud condensation nuclei (CCN), and thus global climate.There is considerable uncertainty about the total radiative

forcing contributed by aerosol particles, mainly from the number concentrations and sizes of CCN (IPCC, 2021). New

particle formation (NPF) is a physical and chemical process, comprising nucleation of gaseous precursors and the

subsequent growth of the nucleated clusters into aerosol particles. NPF has been considered to be an important source of

aerosols and contribute a major fraction to the global CCN budget. Model simulations found that up to 45% of global CCN

were secondary aerosols derived from NPF process, with a large fraction created in the free troposphere (Merikanto et al.,

2009). Furthermore, NPF may be more important to the total CCN budget in the pristine atmosphere than that in the

industrial atmosphere (Gordon et al., 2017).

With the development of technology and instrumentation, NPF events have been observed in diverse atmospheric

environments around the world, including urban, rural, mountain, boreal, and polar regions (Nieminen et al., 2018;

Kerminen et al., 2018; Wang et al., 2017; Peng et al., 2014). High-altitude mountains are particularly interesting

environment for atmospheric NPF studies due to near absence of primary particle sources and long aerosol particle lifetimes

in such environment (Kerminen et al., 2018). Furthermore, the NPF events at the high-altitude sites have been considered as

a potentially important source of particles injected into the free troposphere (Venzac et al., 2008). Unfortunately, there were

few NPF studies at pristine high-altitude sites, mainly due to the challenges in measuring particle numbers and precursors

under the adverse meteorological and topographic conditions such as the extremely low temperature, the thin air and the

steep mountains. Moreover, the parameters and the exact mechanisms of NPF in the limited high-altitude measurements

showed large temporal and spatial variations. For example, variable frequencies of NPF event days were found: 20% during

one year at Jungfraujoch, Switzerland (3580 m a.s.l) (Tröstl et al., 2016a), 56% in spring and 43% in summer at Storm Peak

Laboratory, USA (3210 m a.s.l) (Hallar et al., 2011), 67% during one year in Maïdo observatory, Réunion (2165 m a.s.l)

(Rose et al., 2019), and 80% in post-monsoon season at Mt. Daban, China (3295 m a.s.l) (Du et al., 2015). In Jungfraujoch,

NPF was limited to the availability of highly oxidized organic species, which was restricted by the contact time in the air



masses with the boundary layer (Bianchi et al., 2016). In Storm Peak Laboratory, NPF events were correlated with UV irradiance, but not with $O_3$ concentration and existing particle surface area (i.e. condensation sink) (Hallar et al., 2011). But at Chacaltaya Station in Bolivia, Maïdo observatory, and Jungfraujoch, the NPF frequency was found to be positively

correlated with CS, which was mainly because the increase of transmitted condensable precursors was often accompanied by the increase of CS (Rose et al., 2015; Rose et al., 2019; Boulon et al., 2010). These results indicated the location and seasonal effects of NPF at the high-altitude sites, and more measurements and studies were needed in high-altitude atmosphere.

The Tibetan Plateau (TP) is the largest plateau in China and the highest plateau in the world, with an average altitude

of over 4000 m a.s.l. It is known as the "roof of the world" and "the third pole" of the Earth and plays a fundamental role in the regional climate and environment through various dynamic and thermal effects (Yanai and Wu, 2006). Due to the rare anthropogenic activities, the TP is considered one of the most pristine locations around the world and an ideal location for characterizing remote and regional background aerosols. At the same time, NPF is an assumed key source of CCN production in the TP and has large impact on cloud processes, due to the negligible primary particle emissions. However,

research about NPF in the TP is rather limited. It restrained the understanding of mechanisms and CCN contribution of NPF in the TP, and brought considerable uncertainty about the radiative forcing. Some scientists have conducted the long-term NPF measurement at Himalayan Nepal Climate Observatory at Pyramid (NCO-P) site on the southern TP, and found that the NPF frequency varied in each season with the highest in the monsoon period (Venzac et al., 2008). The occurrence of NPF events in NCO-P was initiated by the up-valley winds which sent the biogenic condensable vapours from Khumbu

valley to the high-altitude location (Bianchi et al., 2021). At Mt. Yulong on the southeastern TP, the NPF frequency was only 14% during pre-monsoon season and the occurrence of NPF events was related to an elevated boundary layer or transported biomass burning pollutants from southern Asia, but not biogenic condensable vapours (Shang et al., 2018). Overall, few NPF studies were conducted in the central TP which can better represent the regional characteristics of TP compared with the southern and southeastern TP. The reason for the variation of NPF frequencies, the mechanisms for

nucleation and growth of nanoparticles, and CCN contribution of NPF in various seasons in the TP remained ambiguous.

In this study, intensive measurements were conducted at Nam Co station (4379 m a.s.l) in the central TP during pre-monsoon and monsoon seasons. Collocated measurements including particle number size distributions (PNSD), trace gases and meteorological parameters, and assisted Weather Research and Forecasting (WRF) and Community Multiscale Air Quality (CMAQ) models were employed to investigate the characteristics of PNSD and NPF. This study aimed to (1)

characterize NPF events in pre-monsoon and summer monsoon season in the central TP, (2) investigate the source of the

NPF events occurrence in pre-monsoon and summer monsoon season in the central TP, and (3) quantify the CCN contribution of NPF in pre-monsoon and summer monsoon season in the central TP.

## 2 Material and methods

### 2.1 Measurement site

An intensive field campaign was carried out at a high mountain observatory at the central Tibetan Plateau, i.e., Nam Co station (30.8°N, 91.0°E; 4730 m a.s.l) during the in-depth study of the atmospheric chemistry over the Tibetan Plateau in the year of 2019, referred as @Tibet 2019 field campaign (Fig. 1). The Nam Co station is located near Nam Co Lake (area: 1920 km$^2$), the highest and largest saltwater lake in the world which is backed by the Nyainqentanglha mountains in the South. The ecology of the surrounding area is semi-arid land dominated by alpine meadow and barren areas. The capital city

(Lhasa) of the Tibet Autonomous Region is about 100 km southeast of the station. The closest town, Dangxiong, is about 70 km southeast of the station, between which are the Nyainqentanglha mountains. Overall, Nam Co station is located in a typical pristine environment and there are almost no local anthropogenic source emissions in this area. The measurement was conducted from 26 April to 22 May, 2019 and 15 June to 25 June, 2019, corresponding to the pre-monsoon season and the summer monsoon season, respectively (Bonasoni et al., 2010; Cong et al., 2015).

### 2.2 Instruments and methods

The sampling was conducted at the observatory field of Nam Co station (Fig. 1c). $O_3$ was detected using ultraviolet (UV) absorption by a "Model 49C Ozone Analyzer" (Thermo Scientific, USA). CO and water content ($H_2O$) were measured by Picarro G2401 based on cavity ring-down spectroscopy. Meteorological parameters including wind speed (WS), wind direction (WD), temperature (T) and relative humidity (RH) were measured by the automatic meteorological station (Met

one Instrument Inc). The photolysis frequencies of $O_3$ ($JO^1D$) were monitored using a spectral radiometer (ultra-fast CCD-detector spectrometer, METCON GmbH, Germany). 99 types of volatile organic compounds (VOC) were measured by an online gas chromatograph coupled with a mass spectrometer and flame ionization detectors (GC-MS/FID) (TH-PKU 300B, Wuhan Tianhong Instrument Co. Ltd., China) in pre-monsoon season (Wang et al., 2014). Black carbon (BC) was measured by Aethalometer (Magee Scientific, model AE33), and the concentration of BC at 880 nm was used in this study

to reduce the influence of Brown carbon (Kirchstetter et al., 2004).

PNSD in the stokes size range of 4nm and 700 nm were obtained by integrating two sets of scanning mobility particle spectrometers (SMPS). The first SMPS measured particles with the size of 4-45 nm, consisting of a TSI Model





3085 DMA and a TSI Model 3776 CPC (with a flow rate of 1.5 L min[-1]). The second SMPS measured particles with the size of 45-700 nm, consisting of a TSI Model 3081 DMA and a TSI Model 3775 CPC (with a flow rate of 0.3 L min[-1]). A silicon diffusion tube was placed before the SMPS, which kept the relative humidity (RH) of the sampling air under 40%. PNSD were corrected for particle losses in the SMPS and the sampling tube, following the method of "equivalent pipe length" as described in Wiedensohler et al. (2012).

Considering that the concentration of VOC in monsoon season has not been measured in this observation, Weather Research and Forecasting (WRF) (version 4.2.1) and The Community Multiscale Air Quality version 5.3.2 (CMAQv5.3.2) models have been adopted to simulate the VOC levels in the surrounding area to assist in the analysis of the role of VOC in NPF events. Similarly, $SO_2$ during pre-monsoon and monsoon was also simulated to help analyze the role of sulfuric acid in NPF events. Detailed information about the model setting and evaluation is provided in Text S1.

To reveal the transport pathway of air masses that arrive at the site, the 48h backward trajectories of the air mass at 600m above the ground (5330 m a.s.l) were computed using the HYSPLIT (Hybrid Single Particle Lagrangian Integrated Trajectory) model and Global Data Assimilation System (GDAS) data.

**2.3 Parameterization of NPF**

In this study, a typical NPF event was defined by the criteria that $PN_{3-10}$ (particle number concentration in the diameter of 3-10 nm) increased obviously, and lasted for more than 2 hours (Fang et al., 2020). The days without particle formation were defined as non-event days. Other days in which the increase of $PN_{3-10}$ was observed but lasted for less than 2 hours were treated as undefined days.

During the NPF events, the formation rate was calculated using the following formula (Cai and Jiang, 2017):

$$J_{d_k} = \frac{dN_{[d_k,d_u]}}{dt} + \sum_{d_g=d_k}^{d_{u-1}} \sum_{d_i=d_{min}}^{+\infty} \beta_{(i,g)} N_{[d_i,d_{i+1}]} N_{[d_g,d_{g+1}]} - \frac{1}{2} \sum_{d_g=d_{min}}^{d_{u-1}} \sum_{d_i^3=\max(d_{min}^3,d_k^3-d_{min}^3)}^{d_{i+1}^3+d_{g+1}^3 \le d_u^3} \beta_{(i,g)} N_{[d_i,d_{i+1}]} N_{[d_g,d_{g+1}]}$$
$$+ n_u \cdot GR_u$$

(1)

Where $J_{d_k}$ is the formation rate of particles at size $d_k$, and $d_u$ is the upper size bound of the target size range. Here $d_k$ and $d_u$ are selected to be 4 and 25 nm, respectively. $N_{[d_k,d_u]}$ is the total number concentration of particles in the diameters of $[d_k,d_u]$. $d_i$ is the lower bound of each measured size bin, and $d_{min}$ is lowest size limit detected by measuring instrument. $\beta_{(i,g)}$ is the coagulation coefficient for the collision between the particle at size of $d_i$ and the particle at size of $d_g$. $n_u$ is the particle size distribution function ($dN/dd_p$). $GR_u$ is the growth rate at size of $d_u$.





The growth rate (GR) was obtained by the mode-fitting method described in Dal Maso et al. (2005). In short, the

PNSD during NPF event days were fitted as the sum of three-mode lognormal distribution. GR was calculated as the

variation of the geometric mean diameter $D_m$ of newly formed mode (4-25 nm) in unit internal (Dal Maso et al., 2005):

$$GR = \frac{\Delta D_m}{\Delta t}$$

(2)

To evaluate the scavenging effects of preexisting particles on condensable vapours, the condensation sink (CS) was

calculated as follow (Dal Maso et al., 2005):

$$CS = 2\pi D \sum \beta_m (D_{p,i}) D_{p,i} N_i$$

(3)

where $D$ is the diffusion coefficient of the condensing vapor, $\beta_m$ is the transition regime correction factor, and $D_{p,i}$

and $N_i$ are the diameter and number concentration in the size class $i$, respectively.

### 2.4 Calculation of CCN concentration

The CCN number concentration was calculated based on the assumption that particles larger than a certain diameter

could act as CCN. The critical diameter ($D_c$) of the CCN activation at the supersaturation ($S_c$) can be calculated based on

$\kappa - K\ddot{o}hler$ theory (Petters and Kreidenweis, 2007):

$$\kappa = \frac{4A^3}{27D_c^3 \ln^2 S_c}$$

(4)

$$A = \frac{4\sigma_{s/a} M_w}{RT \rho_w}$$

(5)

Where $\kappa$ is the hygroscopicity parameter about the composition-dependence of the solution water activity. $\sigma_{s/a}$ is the

water surface tension (0.0728 N m$^{-1}$). $M_w$ and $\rho_w$ are the molecular weight and density of water, respectively. $R$ is the

universal gas constant (J mol$^{-1}$ K$^{-1}$), and $T$ is the absolute temperature (K).

### 3 Results and discussions

### 3.1 Characteristics of meteorology and atmospheric pollutants

The distinct meteorological characterizations were exhibited in the two seasons. As shown in Fig. 2 and Fig. S4,

temperature behavior was characterized by higher value in monsoon season (10.4±4.1 °C) and lower value in pre-monsoon



season (3.1±3.6 °C) with an average value of 5.3±5.1 °C. The relative humidity (RH) seems very similar in the two seasons (50%±21% in pre-monsoon season vs 48%±19% in monsoon season), but the water content ($H_2O$) in the air during monsoon season (1.0%±0.2%) was clearly higher than that in pre-monsoon season (0.6%±0.2%), which reflected the wetter environment in the monsoon period. Wind speed (WS) was comparable during the two seasons, which was 4.2±2.7 m s$^{-1}$ in

pre-monsoon season and 4.5±2.7 m s$^{-1}$ in monsoon season, respectively. Wind direction (WD) showed a clear divergence, with westerly and southwesterly winds prevailing in pre-monsoon season, and southerly winds prevailing in monsoon season (Fig. S5). The frequencies of air masses arriving at the observation station from various directions during the two seasons were shown in Fig. S6. In pre-monsoon season, strong westerlies pass through western Nepal, northwest India and Pakistan (i.e., southern Himalayas). In monsoon season, air masses were derived from Bangladesh and northeast India and

brought moisture that originated in the Bay of Bengal. The seasonality of meteorology was generally in agreement with the previous studies in the TP, which is strongly influenced by the large-scale Asian monsoon circulation (Cong et al., 2015; Bonasoni et al., 2010).

As a background high altitude site on the TP, the Nam Co station displayed a low particle concentration. On average $PM_{0.8}$ was 1.8±1.0 µg m$^3$, which was similar with $PM_1$ (2 µg m$^3$) measured by a high-resolution time-of-flight aerosol mass

spectrometer at Nam Co station in 2015 (Xu et al., 2018a). The particulate matter concentration at Nam Co station was lower than the $PM_1$ observed at Qomolangma Station (4.4 µg m$^3$) on the southern TP and Waliguan Observatory (9.1 µg m$^3$) on the northeastern TP, which resulted from a much longer transport distance of anthropogenic emissions compared with Qomolangma Station and Waliguan Observatory (Zhang et al., 2021). The average concentration of BC was 223±135 ng m$^3$ during the whole measurement period, which was comparable with previous observations (Xu et al., 2018a; Xu et al., 2020;

Wang et al., 2016; Xu et al., 2018b), and represented the background level in the TP region. The average $O_3$ concentration was 58±15 ppbv, which was similar to the results from some high-elevation sites in the TP (Bonasoni et al., 2010; Shang et al., 2018) and higher than the results from Beijing during spring (Chen et al., 2020). Consistent with previous research (Xu et al., 2018a; Cong et al., 2015; Yin et al., 2021), higher PM, BC and $O_3$ concentrations were found during pre-monsoon season (Fig. S7). The higher BC and PM may result from the active biomass-burning emissions in pre-monsoon season on

the southern TP (Cong et al., 2015). The higher $O_3$ in pre-monsoon season were primarily attributed to stratospheric intrusion of ozone (Yin et al., 2021). In contrast, there was no noticeable difference in CO between the two seasons, and the average concentration was 0.12±0.14 ppmv.



### 3.2 Higher frequency of new particle formation in monsoon season

Fig. 2g shows the evolution of PNSD for the entire study. It can be seen that NPF events in the monsoon period were observed almost every day (8 in 10 days, 80%), while the frequency of NPF events was extremely low during the pre-monsoon period (4 in 27 days, 15%). As shown in Table 1, the event frequency at Nam Co station during monsoon season was higher than the result reported at NCO-P site on the southern TP in monsoon season (57%) (Venzac et al., 2008), and similar to the frequency reported at Mt. Daban on the northeastern TP in post-monsoon season (80%) (Du et al., 2015). The frequency at Nam Co station during pre-monsoon season was lower than NCO-P (38%) (Venzac et al., 2008) and comparable with Mt. Yulong on the southeastern TP (14%) (Shang et al., 2018) in the same season. In addition, the NPF frequencies of other high-altitude sites, such as Mt. Tai (Lv et al., 2018), Jungfraujoch (Tröstl et al., 2016a), Maïdo observatory (Rose et al., 2019), Storm Peak Laboratory (Hallar et al., 2011), were basically between those in pre-monsoon and monsoon seasons of Nam Co station. In general, the frequencies of the two seasons at Nam Co station can represent the highest and lowest values of NPF frequencies observed in the TP and other high-altitude sites, and even various environments (such as urban, rural, etc.) (Nieminen et al., 2018).

Table 1 summarized the formation rate ($J$), growth rate (GR), and condensation sink (CS) at Nam Co station and other high-altitude sites. The $J_4$ at Nam Co station varied from 0.38 to 2.43 cm$^{-3}$ s$^{-1}$, with an average value of $1.15\pm0.58$ cm$^{-3}$ s$^{-1}$, which was comparable with the results at Mt. Yulong ($J_3$, 1.18 cm$^{-3}$ s$^{-1}$) (Shang et al., 2018) and Jungfraujoch ($J_{3.2}$, 0.2-7.5 cm$^{-3}$ s$^{-1}$) (Tröstl et al., 2016a). However, the $J$ at Nam Co station was higher than the values at NCO-P site ($J_{10}$, 0.14-0.19 cm$^{-3}$ s$^{-1}$) (Venzac et al., 2008) and Storm Peak Laboratory ($J_8$, 0.39-1.19 cm$^{-3}$ s$^{-1}$) (Hallar et al., 2011), but lower than that at Mt. Tai ($J_3$, 1.33-52.54 cm$^{-3}$ s$^{-1}$) (Lv et al., 2018) and Maïdo observatory ($J_2$, 0.5-30 cm$^{-3}$ s$^{-1}$) (Rose et al., 2019), which may be due to the distinction of particle size range used in the calculation and precursor concentrations between the sites. For example, SO$_2$ at Mt. Tai (3.2 ppb) (Lv et al., 2018) was much higher than that at Mt. Yulong (0.06 ppb) (Shang et al., 2018). The average GR in the size range of 4-25 nm at Nam Co station varied from 1.5 to 5.6 nm h$^{-1}$, with the average value of 4.0 $\pm1.2$ nm h$^{-1}$, which was comparable with those at Mt. Yulong (3.2 nm h$^{-1}$) (Shang et al., 2018) and Jungfraujoch (4.0 nm h$^{-1}$) (Tröstl et al., 2016a), and higher than those at NCO-P site ($1.8\pm0.7$ nm h$^{-1}$) (Venzac et al., 2008) and Mt. Daban (2.0 nm h$^{-1}$) (Du et al., 2015), but lower than that at Maïdo observatory (8-45 nm h$^{-1}$) (Rose et al., 2019). The average CS was $0.15\times10^{-2}\pm0.07\times10^{-2}$ s$^{-1}$ at Nam Co station, which was comparable with those at Mt. Yulong ($0.2\times10^{-2}$ s$^{-1}$) (Shang et al., 2018), Maïdo observatory ($0.02\times10^{-2}$-$2\times10^{-2}$ s$^{-1}$) (Rose et al., 2019) and Storm Peak Laboratory ($0.12\times10^{-2}$ s$^{-1}$) (Hallar et al., 2011), and much lower than that at Mt. Tai ($1.1\times10^{-2}$-$4.8\times10^{-2}$ s$^{-1}$) (Lv et al., 2018). In a word, the $J$, GR, and CS at Nam Co station were within average levels in the high-altitude sites. Meanwhile, there was no significant variation in $J$, GR

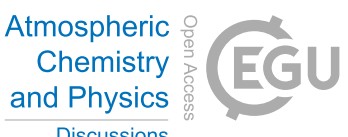

and CS between pre-monsoon and monsoon seasons, although the NPF frequencies of the two seasons were quite different. The key factors determining the occurrence of NPF events need to be further studied.

### 3.3 Frequent NPF events driven by organic matter brought by summer monsoon

Whether an NPF event can occur was mainly related to 1) the CS, which mainly referred to the scavenging rate of precursors, clusters, and newly formed particles by background aerosols. High CS can lead to the continual reduction in newly formed particle number concentration, and inhibit the occurrence of NPF; 2) the gaseous precursors that can participate in nucleation and growth, including sulfuric acid (Kulmala et al., 2013), dimethylamine (Yao et al., 2018), ammonia (Xiao et al., 2015) and VOC (Tröstl et al., 2016b; Fang et al., 2020; Qiao et al., 2021). A sufficiently high

concentration of low volatility vapors (precursors) can contribute to persistent nucleation and generating new atmospheric particles; 3) meteorological factors including WD, RH, temperature, etc, which can influence the occurrence and intensity of NPF events by directly or indirectly affecting the source and sink parameters.

    In the following, the discussion will focus on the key factors for the occurrence of NPF events described above. For clarity, NPF events days occurring in pre-monsoon and monsoon seasons were named NPF-pre days and NPF-monsoon

days, respectively. Non-events days in pre-monsoon season were termed non-event days, as non-event days in monsoon season were scarce.

### 3.3.1 Condensation sink

    As shown in Fig. 3a, the levels of CS in NPF-pre days, NPF-monsoon days and non-event days were similar during 11:00–18:00 (the occurrence time of NPF events), although the CS in the early morning of NPF-pre days seems to be

slightly lower. The CS was mainly in the range of $0.1 \times 10^{-2}$-$0.15 \times 10^{-2}$ $s^{-1}$ during the NPF occurrence time, which was much lower than that at most locations in China, such as ~0.01 $s^{-1}$ in urban Beijing (Deng et al., 2021), and $0.1 \times 10^{-2}$ to $28.4 \times 10^{-2}$ $s^{-1}$ at Mt. Tai (Lv et al., 2018), and comparable with that at Mt. Yulong (~$0.2 \times 10^{-2}$ $s^{-1}$) (Shang et al., 2018). The result varied from previous studies which reported much lower CS during NPF days than that in non-event days (Zhou et al., 2021; Lv et al., 2018). It indicated that the occurrence of NPF events at Nam Co station was not influenced by the lower CS.

The similar phenomenon was found at Mt. Yulong (Shang et al., 2018). In addition, previous studies found that CS was on average positively correlated with the occurrence of NPF events in some high-altitude observations at a larger scale, such as Maïdo observatory and Jungfraujoch (Rose et al., 2019; Boulon et al., 2010). When the gaseous precursors were transported to the observation site to trigger NPF events, the pre-existing particles were transported simultaneously and caused the



increase in CS. Although the topography and the rapidly changing conditions related to complex atmospheric dynamics at

Nam Co station were similar with Maïdo observatory and Jungfraujoch, the relationship between CS and the occurrence of

NPF events seemed different. Whatever, the result indicated that CS was not the decisive factor controlling the occurrence

of NPF events at Nam Co station.

### 3.3.2 Gas precursors

Gaseous sulfuric acid has been identified as the key precursor for nucleation and initial growth due to its low volatility

(Kulmala et al., 2013; Qiao et al., 2021), and the photochemical formation of $SO_2$ was the major source for sulfuric acid in

the atmosphere. However, the level of $SO_2$, the photochemical oxidation intensity of $SO_2$, and CS were comparable between

NPF days and non-event days, which indicated that sulfuric acid may not govern the occurrence of NPF events at Nam Co

station. Here, the low levels of $SO_2$ ($0.03\pm0.03$ ppb) in NPF days and non-event days were present in Fig. S8. The

photochemical oxidation intensity of $SO_2$ was indicated by J ($O^1D$). As shown in Fig. 3b, NPF-pre days, NPF-monsoon

days and non-event days shared the same level of J ($O^1D$) ($3 \times 10^{-2}$-$6 \times 10^{-2}$ $s^{-1}$) during the occurrence time of NPF events.

In addition to sulfuric acid, organics were also considered to be an important factor of NPF events. Observations and

laboratory experiments have found that organics may participate in or even dominate the nucleation and growth process in

NPF events in pristine environments and the preindustrial atmosphere. For example, CLOUD (Cosmics Leaving Outdoor

Droplets) experiments observed obvious NPF events from highly oxidized organics without the involvement of sulfuric acid

(Kirkby et al., 2016). At the high-altitude sites of Jungfraujoch and Himalaya, NPF events occurred mainly through the

condensation of highly oxygenated molecules (HOMs) (Bianchi et al., 2016; Bianchi et al., 2021). Due to instrument status,

VOC measurement was only available in pre-monsoon season. The concentration of TVOC (total VOC) showed a

noticeably higher value (20%) during 11:00-18:00 on NPF-pre days compared with non-event days (Fig. 3c). Aromatics,

which can be used as the indicator of anthropogenic emissions, also exhibited a higher level (20%) during NPF-pre days

(Fig. 3d). This suggested that VOC may be the key factor in determining the occurrence of NPF events. In order to further

evaluate the role of VOC, we used WRF/CMAQ models to simulate the spatial distribution of VOC concentration in

pre-monsoon and monsoon seasons. It can be found that the average VOC concentration of NPF-monsoon days was about 2

ppb higher than that of NPF-pre days, and about 4 ppb higher than that of non-event days at Nam Co station (Fig. 4). In

addition, one recent research has found that the concentration of monoterpene-derived HOMs in East Asia was higher in

summer (June-August) than that in Spring (March-May) by using GEOS-Chem global chemical transport model (Xu et al.,

2022). All the results indicated that the frequent NPF events in monsoon season were caused by the higher concentration of





organic precursors. And pure organic nucleation may be the dominant NPF mechanism at Nam Co station.

### 3.3.3 Meteorology

While the concentrations of organic precursors have explained the notable distinction in NPF frequencies in pre-monsoon season and monsoon season, the external factors driving the apparent difference in VOC levels between the two seasons and other conditions that may affect the characteristics of NPF were still unknown. This indicated that a further investigation into other NPF-related variables was still required.

WD can reflect the local situation for air mass and the source of air pollutants. The wind rose plots of non-event days, NPF-pre days and NPF-monsoon days were present in Fig. 5. It can be found that the dominant WD on non-event days was 285 westerly and the WS was mostly below 10 m/s. In comparison, WD on NPF-pre days and NPF-monsoon days mainly come from the South, mostly above 8 m/s. Similar patterns can be found from the time series (Fig. S2), that was, the WD from 3 May to 5 May (NPF-pre days) resembled that during summer monsoon, mainly from the South, and the overall WS was relatively high. The example of the spatial distribution of wind in non-event days, NPF-pre days and NPF-monsoon days showed the same phenomenon on a larger scale (Fig. S9). These results suggested that when the southerly wind with high 290 WD occurred, it may bring organic precursors from the southern region (northeast India) to Nam Co station, driving the occurrence of NPF events in this area. This explained the phenomenon of the higher VOC concentration in NPF days than non-event days. The frequent NPF events in monsoon season resulted from the higher concentration of organic vapours brought by the frequent southerly wind (summer monsoon). This conclusion was confirmed by the more frequent air masses from the South during monsoon season (Fig. S6). Similar result was found in the recent study which showed that the Indian 295 summer monsoon acted as a facilitator for transporting gaseous pollutants (Yin et al., 2021).

In Fig. 6, we showed the diurnal variations of meteorological factors during NPF-pre days, NPF-monsoon days and non-event days at Nam Co station. First, the temperature in NPF-pre days and non-event days were generally close, but the temperature of NPF-monsoon days was obviously higher. The similar temperature in NPF-pre days and non-event days suggested that temperature was not a crucial factor for NPF event occurrence. Previous studies have found that higher 300 temperature can increase the formation rate of HOMs, but reduce the volatility of HOMs in the meanwhile. The effect of temperature on the nucleation rate may not be important in a limited ambient temperature range (Stolzenburg et al., 2018). But the higher temperature during NPF-monsoon days may promote monoterpene emissions, which favored particle nucleation and growth (Andreae et al., 2022). This may explain the phenomenon for the higher VOC concentration in NPF-pre days than NPF-monsoon days. Second, WS shared the same level during 11:00-18:00 on NPF days and non-event

days, while WS before the occurrence of NPF events in NPF days was higher than that of non-event days. Third, the

average RH was similar in both NPF days and non-event days, although NPF-pre days had a wide range of RH changes.

Laboratory studies have shown that water vapour can suppress the formation of extremely low volatility organic compounds

(ELVOC) from monoterpenes (Bonn et al., 2002). However, our result was the opposite. During 11:00-18:00, the water

content ($H_2O$) in the air during NPF-monsoon days was the highest, NPF-pre days was the second, and non-event days was

the lowest. This indicated the inhibition effect of water vapour was not important at Nam Co station. The similar finding

was also observed at remote sites in the subboreal forest of North America (Andreae et al., 2022).

     In all, the level of organic vapours governed the occurrence of NPF events at Nam Co station, while condensation sink

and gaseous sulfuric acid had no effect. The frequent NPF events in summer monsoon season resulted from the higher

concentration of organic vapours, which was brought from northeast India by the strong southerly monsoon.

**3.4 Significant increase of CCN in monsoon season**

     The CCN concentration was estimated following the method introduced in Sect. 2.4. The hygroscopicity parameter κ

was assumed to be a constant value of 0.12 throughout the measurement period, according to the previous measurement of

chemical composition in TP (Shang et al., 2018). As a result, the $D_c$ at $S_c$ levels of 0.6% and 1.2% were $73.4\pm1.3$ and

$46.3\pm0.8$ nm, respectively. No noticeable difference in $D_c$ was found between pre-monsoon and monsoon seasons ($S_c =$

0.6%: $74.4\pm1.0$ vs $72.4\pm1.1$ nm, $S_c = 1.2\%$: $46.9\pm0.6$ vs $45.7\pm0.7$ nm). There could be uncertainties in the values of

κ and $D_c$, but they had little impact on the final result of CCN concentration (Shang et al., 2018).

     The aerosol production and CCN production during an NPF event can be obtained by comparing the particle number

concentration at the beginning of the increase of the target particles ($N_{init}$) with the maximum number concentration ($N_{max}$),

as introduced by Rose et al. (2017). The $N_{init}$ and $N_{max}$ are hourly average concentrations as shown in Fig. S10. Higher daily

aerosol production and CCN production were observed during monsoon season compared with that in pre-monsoon season

(Fig.7). In monsoon season, Nucleation-mode particles (4-25 nm) started to increase quickly at around 11:00 when

nucleation occurred. The freshly nucleated particles grew to larger sizes due to condensation and coagulation of the

pre-existing particles within several hours, which contributed to the increase of Aitken-mode particles (25-100 nm) from

15:00. The average daily aerosol production of Nucleation-mode particles and Aitken-mode particles in monsoon season

was around 3400 $cm^{-3}$ and 1200 $cm^{-3}$, respectively, and the enhancement factors (EFs, $N_{max}/N_{init}$) were 9.2 and 3.6,

respectively. As for CCN, the average production of CCN at $S_c$ of 0.6% and 1.2% in monsoon season was 180 and 518

$cm^{-3}$, respectively, and the EFs were 1.7 and 2.2, respectively. The EFs of CCN were lower than previous studies because





they only considered NPF days (Rose et al., 2017; Shen et al., 2016). Although the particles and CCN at around 11:00 in pre-monsoon season were comparable with monsoon season, the production of aerosols and CCN was much lower. The

average daily aerosol production of Nucleation-mode particles and Aitken-mode particles in pre-monsoon season were around 500 cm$^{-3}$ and 300 cm$^{-3}$, respectively. And the CCN production at $S_c$ of 0.6% and 1.2% was 170 and 286 cm$^{-3}$, respectively. The average daily production of Nucleation-mode particles, Aitken-mode particles and CCN at $S_c$ of 1.2% in monsoon season was 5.8, 3 and 0.8 times higher than that in pre-monsoon season, respectively.

The high-frequency NPF events in the summer monsoon period markedly increased the number concentration of

atmospheric aerosols. The average PNSD during pre-monsoon and monsoon seasons were plotted in Fig. 8a with much higher number concentrations occurring during monsoon season. The mean total particle number concentration (PN$_{4-700}$) in monsoon season was $3647\pm2671$ cm$^{-3}$, which was more than 2 times higher than that of pre-monsoon season ($1163\pm1026$ cm$^{-3}$). Although the measured particle size ranges were not the same with other studies, the results can still be comparable as the background particles were mainly distributed in tens to hundreds of nanometers. As shown in Table 1, PN$_{4-700}$ at Nam

Co station in pre-monsoon season was comparable with other high-altitude sites around the world, while PN$_{4-700}$ in monsoon season was much higher due to the frequent NPF events. The atmospheric particles contributed by new particle nucleation and growth in monsoon season were mainly concentrated below 100 nm. Among them, the concentration of Nucleation-mode particles in monsoon season was about 2 times higher than that in pre-monsoon season, and the concentration of Aitken-mode particles was 3.5 times higher than that in pre-monsoon season. In contrast,

Accumulation-mode particles (>100 nm) which were related to the secondary formation process and long-range transport (Wang et al., 2013; Vu et al., 2015), were nearly at the same level in the two seasons (around 200 cm$^{-3}$). As shown in Fig. 9, the average number concentrations of CCN at $S_c$ of 0.6% and 1.2% in monsoon season were $434\pm242$ and $863\pm628$ cm$^{-3}$, respectively. The results were 10% and 56% higher than those in pre-monsoon season at $S_c$ of 0.6% ($396\pm177$ cm$^{-3}$) and 1.2% ($552\pm261$ cm$^{-3}$).

It should be noted that the CCN concentration and production we estimated above were local ground levels, which can be considered as the minimum values. That is because those smaller particles (<40 nm) formed by NPF may further grow and reach CCN size in the following days during transportation, as the coagulation sink affecting the lifetime of aerosols was at a low level. This mechanism was verified at different high-altitude stations such as NCO-P site, Chacaltaya and Jungfraujoch (Venzac et al., 2008; Rose et al., 2015; Tröstl et al., 2016a). The Nucleation-mode and Aitken-mode particle

bands in the morning reflected the continuation of NPF events. As a result, the amount of potential CCN in monsoon season could be much larger than the result measured here, due to the high level of the smaller particles. Therefore, the climate

effect from aerosols may have a large variance between pre-monsoon and monsoon season.

## 4 Summary and conclusions

The PNSD, trace gases and meteorological parameters were measured at Nam Co station (4379 m a.s.l) in the central Tibetan Plateau (TP) during pre-monsoon and summer monsoon seasons. Firstly, meteorological conditions between pre-monsoon and monsoon seasons were distinct with higher temperature and water content in monsoon season. Strong westerlies pass through western Nepal, northwest India and Pakistan in pre-monsoon season, while air masses were mainly derived from the South (Bangladesh and northeast India) in the monsoon period. Secondly, as the pristine high-altitude site, low levels of particles and gaseous pollutants were displayed at Nam Co station. Relatively higher pollutant concentration was found during pre-monsoon season due to less favorable atmospheric circulation.

The most important finding of this study was that there was an evident distinction in the frequencies of NPF events at Nam Co station with 15% in pre-monsoon season and 80% in monsoon season, which can represent the lowest and highest values of NPF frequencies observed in the TP and other high-altitude sites, even around the world. In addition, there was no noticeable variation in $J$, GR, and CS between pre-monsoon and monsoon seasons at Nam Co station. Through the comprehensive analysis of the measured CS, gaseous precursors and meteorological conditions, supplemented by the model simulations, we found that the level of organic vapours governed the occurrence of NPF events, while condensation sink and gaseous sulfuric acid had no effect. The frequent NPF events in monsoon season resulted from the higher concentration of organic vapours, which was brought from northeast India by the strong southerly monsoon.

The frequent NPF events in the summer monsoon season had elevated the number concentration of atmospheric aerosols and CCN at Nam Co station. The average daily production of Nucleation-mode particles, Aitken-mode particles and CCN at $S_c$ of 1.2% in monsoon season was 5.8, 3 and 0.8 times higher than that in pre-monsoon season, respectively. The average total particle number concentration in monsoon season was more than 2 times higher than that in pre-monsoon season, mainly contributed by Nucleation-mode and Aitken-mode particles. And the average number concentration of CCN in monsoon season was 10%-56% higher than that in pre-monsoon season. Considering that the smaller particles (< 40 nm) formed by NPF may further grow and reach CCN size during the following days due to the low-level coagulation sink, the amount of potential CCN in monsoon season could be much larger than our local measurement results. It may markedly affect the earth's radiation balance and global climate. Our result emphasized the role of organic in the occurrence of NPF events in the TP. Seasonal effect of NPF should be considered in model simulations when calculating the amount of aerosols and CCN in high-altitude atmosphere.

*Data availability.* The data provided in this paper can be obtained from the author upon request (minhu@pku.edu.cn).

*Supplement.* An independent supplement document is available.

*Authorship contributions.* Lizi Tang: Investigation, Data curation, Methodology, Formal analysis, Writing - original draft,
Writing - review & editing. Min Hu: Project administration, Supervision, Funding acquisition, Writing - review & editing.
Dongjie Shang: Investigation, Data curation, Methodology, Formal analysis. Xin Fang: Investigation, Data curation,
Methodology, Formal analysis. Jianjiong Mao: Investigation, Data curation. Wanyun Xu: Data curation. Jiacheng Zhou:
Data curation. Weixiong Zhao: Data curation. Yaru Wang: Data curation. Chong Zhang: Data curation. Yingjie Zhang: Data
curation. Jianlin Hu: Data curation. Limin Zeng: Data curation. Chunxiang Ye: Project administration, Funding acquisition,
Data curation. Song Guo: Writing - review & editing. Zhijun Wu: Writing - review & editing.

*Competing interests.* The authors declare that they have no conflict of interest.

*Acknowedgements.* The research has been supported by the National Natural Science Foundation of China (91844301 and
91544214), National Research Program for Key Issues in Air Pollution Control (DQGG0103), National Key Research and
Development Program of China (No. 2016YFC0202000: Task 3), and the second Tibetan Plateau Scientific Expedition and
Research Program (STEP, 2019QZKK0601).





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





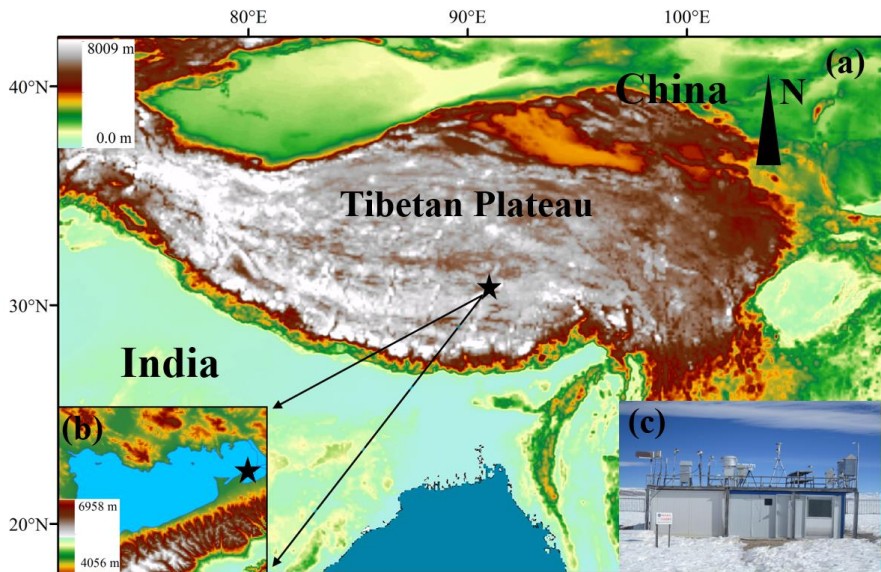

**Figure 1.** Location map for (a) the Tibetan Plateau, (b) Nam Co station, colored according to altitude. (c) The appearance

drawing of atmospheric environment observatory field of Nam Co station (Yin et al., 2020).


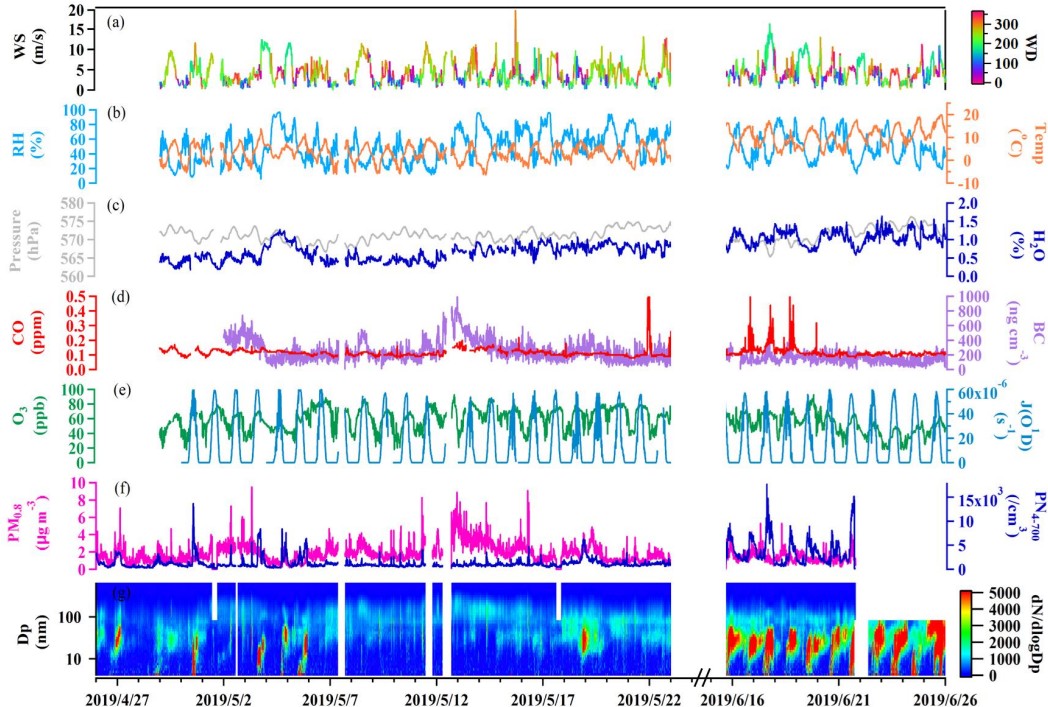

**Figure 2.** Time series during pre-monsoon and monsoon: (a) the wind speed and wind direction, (b) the ambient

temperature and the relative humidity (RH), (c) the pressure and water content ($H_2O$) in air, (d) CO concentration and the

black carbon (BC) concentration, (e) O3 concentration and the photolysis frequencies of $O_3$ ($JO^1D$), (f) the $PM_{0.8}$ mass

concentration and the number concentration of particles in size of 4-700 nm ($PN_{4-700}$), (g) the particle number size

distribution (PNSD).






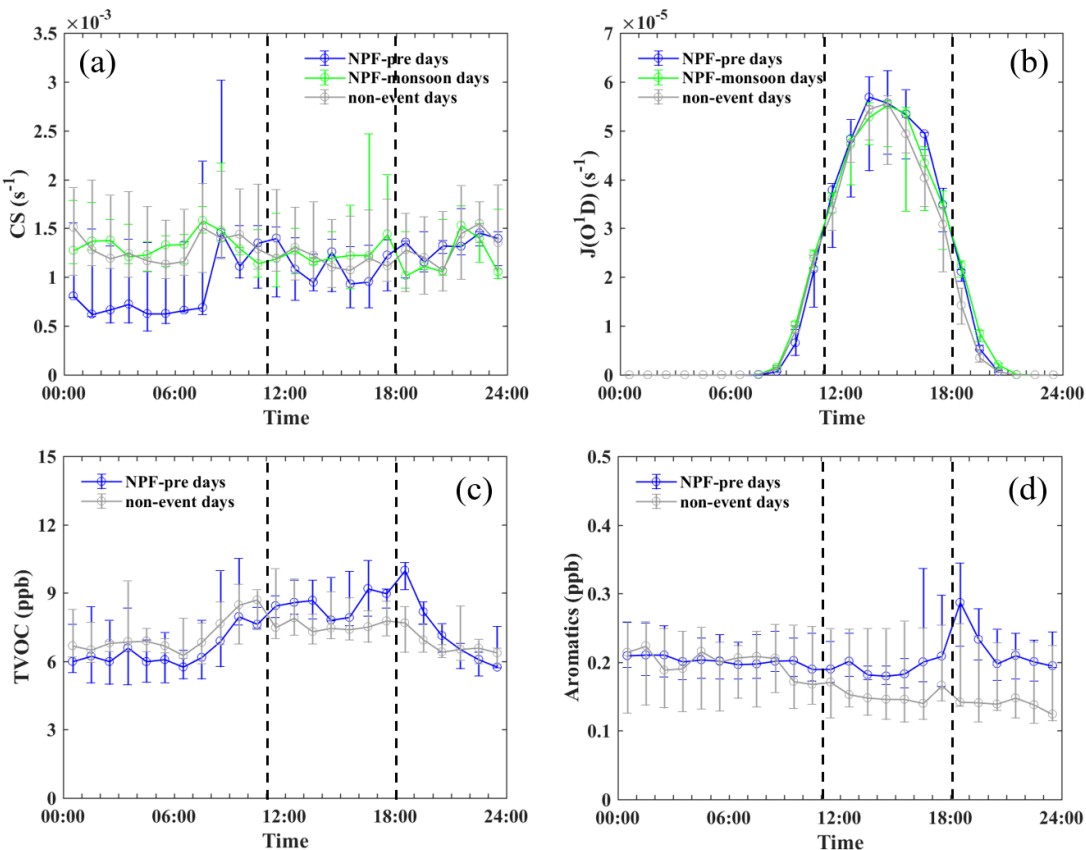


**Figure 3.** Diurnal variations of (a) condensation sink (CS), (b) JO[1]D, the total concentration of (c) VOC (TVOC) and (d)

aromatics in NPF-pre days, NPF-monsoon days and non-event days.






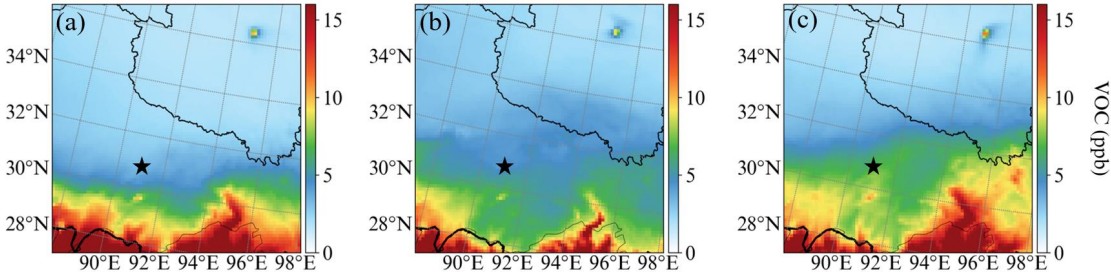

**Figure 4.** Model spatial distribution of TVOC during (a) non-event days, (b) NPF-pre days, and (c) NPF-monsoon days.

The star is Nam Co station.


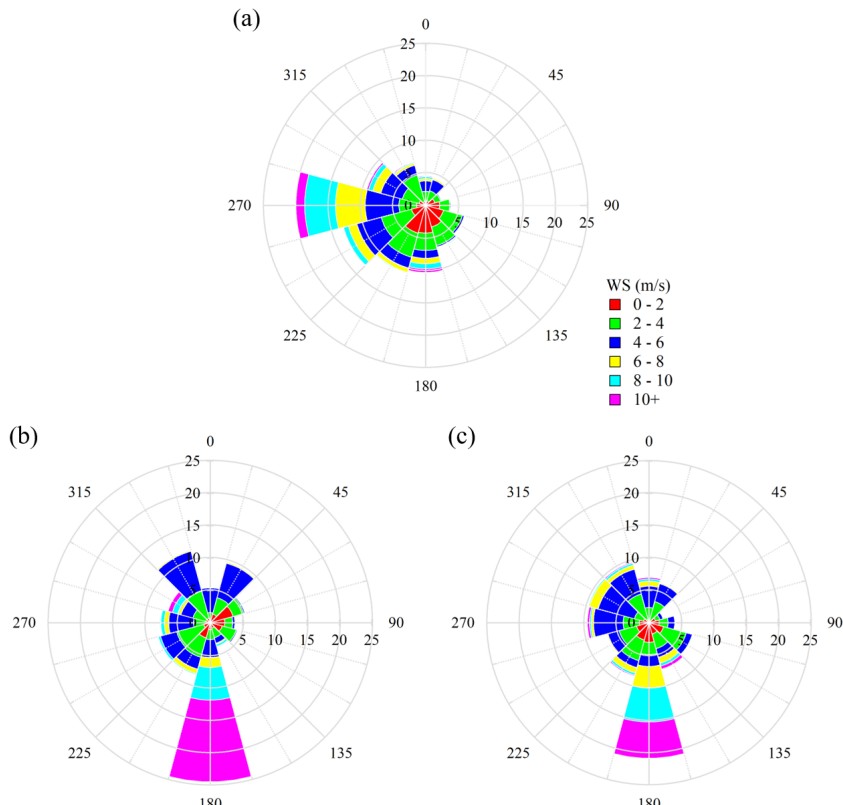

**Figure 5.** Wind rose plots of (a) non-event days, (b) NPF-pre days and (c) NPF-monsoon days. The length of each spoke on

the circle represents the probability of wind coming from a particular direction at a certain range of wind speed.




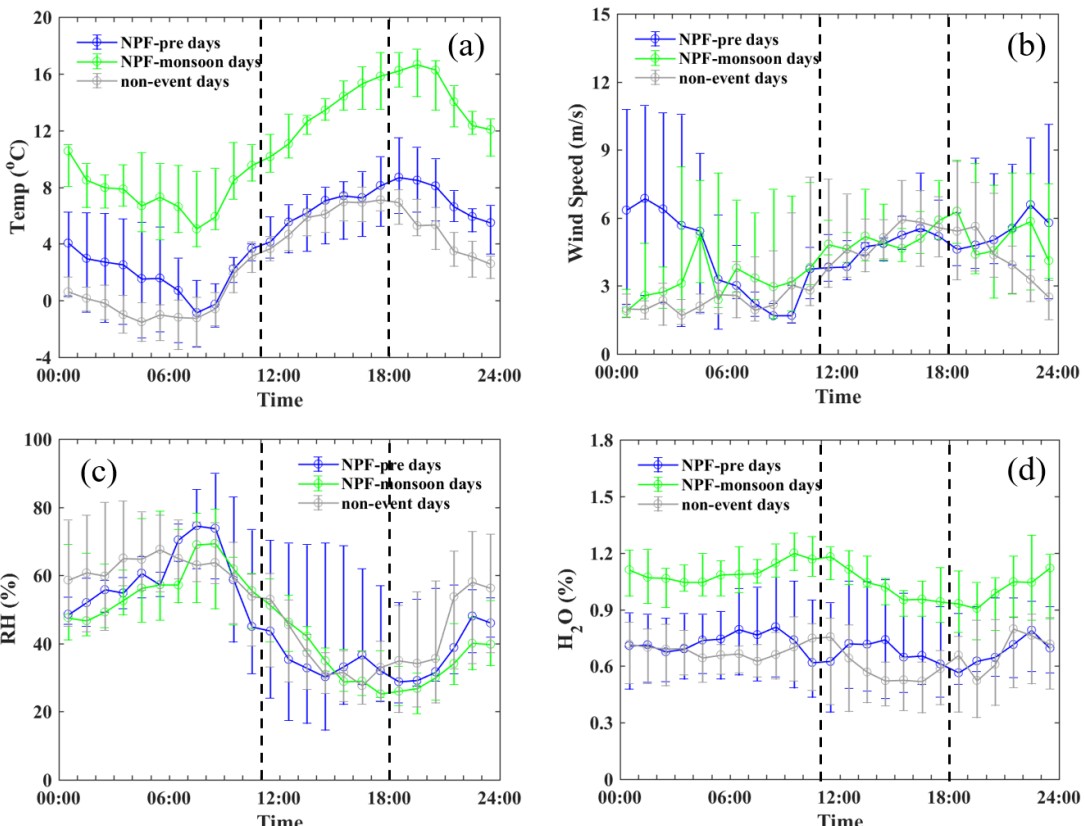

**Figure 6.** Diurnal variations of (a) temperature, (b) wind speed, (c) RH and (d) the water content ($H_2O$) in the air in NPF-pre days, NPF-monsoon days and non-event days.






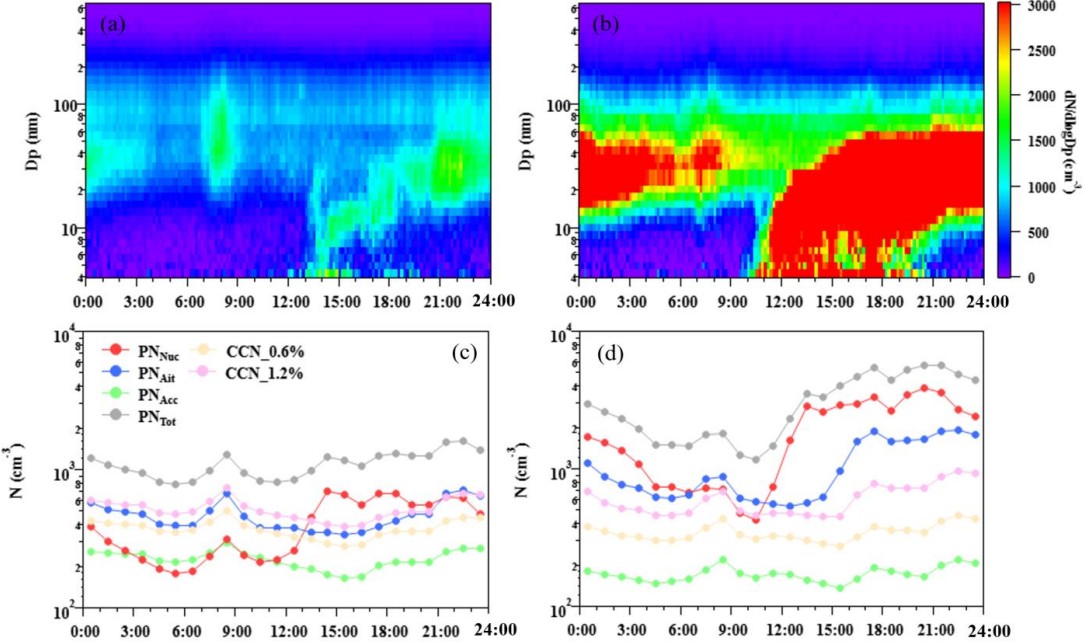

**Figure 7.** Diurnal variations of particle number size distributions (PNSD) and the number concentrations of Nucleation-mode particles (PN$_{Nuc}$), Aitken-mode particles (PN$_{Ait}$), Accumulation-mode particles (PN$_{Acc}$), the total particles (PN$_{Tot}$), CCN at $S_c$ of 0.6% (CCN_0.6%) and 1.2% (CCN_1.2%) during pre-monsoon (a and c) and monsoon (b and d) seasons.





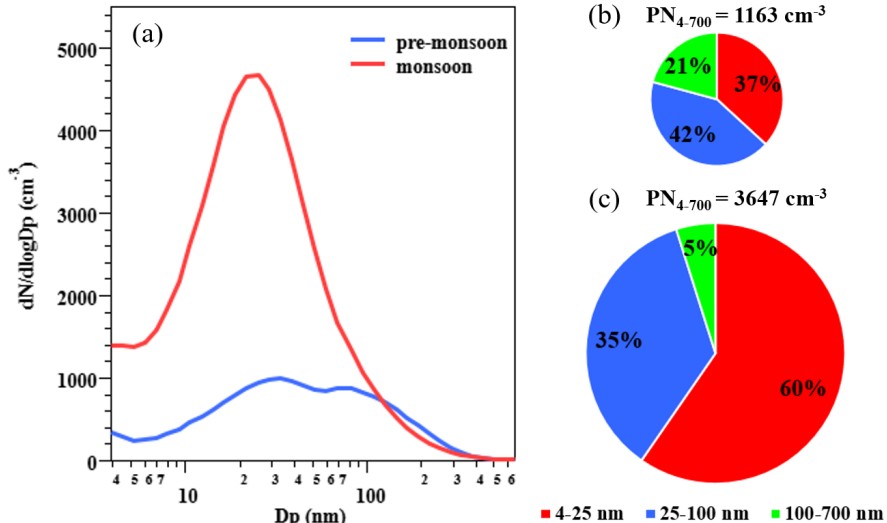


**Figure 8.** Particle number size distribution (PNSD) at Nam Co station. (a) Mean PNSD in pre-monsoon and monsoon; the contribution of different size ranges (4-25 nm, 25-100 nm, 100-700 nm) to the total particle number concentration (b) in pre-monsoon and (c) in monsoon.






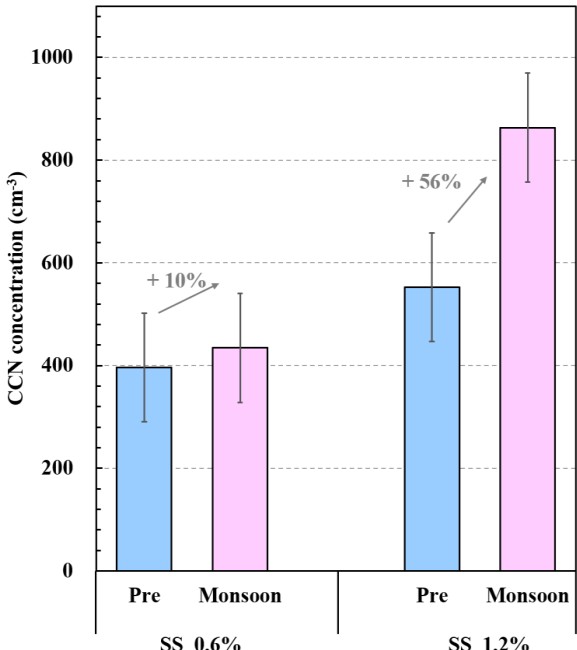

**Figure 9.** Mean number concentration of CCN at $S_c$ of 0.6% and 1.2% in pre-monsoon and monsoon seasons.






**Table 1.** Comparison of particle number concentration (PN) and NPF parameters (NPF frequency, CS, J, GR) with other high-altitude sites around the world

| Site | Altitude (m) | Observation Date and season[a] | PN (cm$^{-3}$) | NPF frequency | CS×10$^{-2}$ (s$^{-1}$) | $J$[b] (cm$^{-3}$ s$^{-1}$) | GR[c] (nm h$^{-1}$) | Reference |
|---|---|---|---|---|---|---|---|---|
| Nam Co station, China | 4379 | May 2019, pre-monsoon | 1163±1026 (PN$_{3-700}$) | 15% | 0.14±0.07 | 1.11±0.79 ($J_4$) | 4.2±0.9 (GR$_{4-25}$) | This study |
| | | June 2019 monsoon | 3647±2671 (PN$_{3-700}$) | 80% | 0.15±0.05 | 1.08±0.21 ($J_4$) | 3.8±0.8 (GR$_{4-25}$) | |
| NCO-P[d], Nepal | 5079 | April-June 2007, pre-monsoon | NA | 38% | NA | 0.14 ($J_{10}$) | 1.8±0.7 | Venzac et al. (2008) |
| | | July-September 2007, monsoon | NA | 57% | NA | 0.19 ($J_{10}$) | NA | |
| Mt. Daban, China | 3295 | September-October 2013, post-monsoon | 2400 (PN$_{12-478}$) | 80% | NA | NA | 2.0 | Du et al. (2015) |
| Mt. Yulong, China | 3410 | May–April 2015, pre-monsoon | 1600±1290 (PN$_{3-10000}$) | 14% | 0.2 | 1.18 ($J_3$) | 3.2 | Shang et al. (2018) |
| Mt. Tai, China | 1534 | July-August 2014, summer | NA | 21% | 1.1-4.8 | 1.33-52.54 ($J_3$) | 1.15-7.76 (GR$_{3-20}$) | Lv et al. (2018) |
| Jungfraujoch, Switzerland | 3580 | July 2013-June 2014, one year | 200-1000 (PN$_{10-600}$) | 20% | NA | 0.2-7.5 ($J_{3.2}$) | 4±2.3 (GR$_{5-15}$) | Tröstl et al. (2016a) |
| Maïdo observator, Réunion | 2165 | 2015, one year | NA | 67% | 0.02-2 | 0.5-30 ($J_2$) | 8-45 (GR$_{12-19}$) | Rose et al. (2019) |
| Storm Peak Laboratory, USA | 3210 | March-May 2001-2009, spring | NA | 56% | 0.12±0.05 | 0.39±0.05 ($J_8$) | 7.5±4.5 | Hallar et al. (2011) |
| | | June-July 2001-2009, summer | NA | 43% | 0.13±0.06 | 1.19±0.34 ($J_8$) | 9.1±6.9 | |

[a] The seasons in Tibet measurements were divided into pre-monsoon, monsoon and post-monsoon seasons. The seasons not in Tibet measurements were divided into spring, summer, autumn and winter.

[b] The subscript indicated the $J$ of particles at the certain size.

[c] The subscript indicated the particle size range used for calculation of GR. No explanation meant that it was not specified in the literatures.

[d] Nepal Climate Observatory at Pyramid site



NA: Data not found in the literatures