# Peer review of "High frequency of new particle formation events driven by summer monsoon in the central Tibetan Plateau, China"

_Atmospheric Chemistry and Physics, 2022_

## Author Comment (AC1)

**Reply to comments**

Journal: Atmospheric Chemistry and Physics

Manuscript Number: acp-2022-440

Title: "High frequency of new particle formation events driven by summer monsoon in the central Tibetan Plateau, China"

Author(s): Lizi Tang, Min Hu, Dongjie Shang, Xin Fang, Jianjiong Mao, Wanyun Xu, Jiacheng Zhou, Weixiong Zhao, Yaru Wang, Chong Zhang, Yingjie Zhang, Jianlin Hu, Limin Zeng, Chunxiang Ye, Song Guo, Zhijun Wu

**I. Reply to Reviewer 1**

**Reply to Reviewer 1's overall comments:**

*This manuscript investigates atmospheric new particle formation (NPF) taking place in the central Tibetan Plateau. To my knowledge, there are no prior publications of NPF in this location, so the obtained results can be considered worth publishing. The paper itself if well organized and the conducted analysis appears to be scientifically sound. There are, however, two major issues that require further consideration.*

> We appreciate the comments from the reviewer on this manuscript. We have answered them point to point in the following paragraphs (the texts italicized are the comments, the texts indented are the responses, and the texts in blue are revised parts in new manuscript). In addition, all changes made are marked in the revised manuscript. Thanks for the reviewer's affirmation on our work.

**Reply to Reviewer 1's major comments (2):**

*1. First, the measurements periods are rather short, about 4 weeks for the pre-monsoon season and less than 2 weeks for the monsoon season. As a result, it remains unclear how representative the obtained results are for this location during these two seasons.*

> Thanks for the comment. The measurements periods were a little limited as the reviewer described. But our measurements periods can be representative for this location during pre-monsoon season and monsoon season as follows:
>
> 1) The intensity of Indian Summer Monsoon during the two measurements periods can represent that in the whole pre-monsoon and monsoon seasons, respectively. The intensity of Indian Summer Monsoon is an important indicator to distinguish the monsoon season. Here the intensity of Indian Summer Monsoon (ISM) was indicated by the ISM Index, which are defined by the negative outgoing longwave radiation anomalies (with respect to the climatological annual cycle) averaged over the Bay of Bengal–India region (10°–25°N, 70°–100°E) (Wang and Fan, 1999). As shown in Fig. R1, the measurement periods (green boxes) were in the pre-monsoon season (March-May) and monsoon season (June-September), respectively. And the IMS index during the two measurements periods were equivalent to those of the whole pre-monsoon season (average: -19.5 vs -20.7 W m$^{-2}$) and monsoon season (average: 27.0 vs 26.3

W m$^{-2}$), respectively. Therefore, we considered that these two observation periods are representative in the seasonal characteristics in pre-monsoon season and monsoon season, respectively.

[Figure]

**Figure R1.** The Indian Summer Monsoon (ISM) Index in 2019. The measurements periods are marked by the green boxes.

2) The characteristics of meteorology and atmospheric pollutants in the two measurements periods was generally in agreement with the previous long-term studies at Nam Co station and other sites in the Tibetan

Plateau (TP) (Yin et al., 2017; Cong et al., 2015; Bonasoni et al., 2010; Xu et al., 2018). Both previous research and this study showed that, strong westerlies pass through western Nepal, northwest India and

Pakistan in pre-monsoon season, while air masses were mainly derived from Bangladesh and northeast

India and brought moisture that originated in the Bay of Bengal in monsoon season (Fig. R2) (Yin et al.,

2017). The temperature, WS and RH in the two measurements periods were matched with those in the whole pre-monsoon and monsoon season of 2020 at Nam Co station (Fig. R3) (National Tibetan Plateau

Data Center). The average temperature in pre-monsoon and monsoon seasons were around 3 and 10 °C, respectively. WS showed no difference between pre-monsoon and monsoon seasons with the average value of 4 m/s. The average level of RH was similar between the two seasons, but the variation range of

RH was larger in pre-monsoon season. In addition, the level of PM, BC and ozone in the two measurements periods were matched with those in the whole pre-monsoon and monsoon season at the other site of the TP (Fig. R4) (Bonasoni et al., 2010; Xu et al., 2018). The average concentrations of $PM_1$

($PM_{0.8}$) in pre-monsoon and monsoon seasons were around 1 and 2 $\mu g/m^3$, respectively. The concentration of BC was at a level of hundreds nanogram per cubic meter in pre-monsoon season, while at a lower level in monsoon season. Ozone showed the similar pattern with BC. It should be noted that there will be some differences between this study and other results due to the differences in time resolution. In a word, the characteristics of meteorology and atmospheric pollutants in the two measurements periods in this study can well reflect those in pre-monsoon and monsoon seasons.

[Figure]

**Figure R2**. Comparison of trajectories between this study and previous study at Nam Co station.

Backward HYSPLIT trajectories for each measurement day (black lines in the maps), and mean back trajectory for six HYSPLIT clusters (colored lines in the maps) arriving at Nam Co Station in spring (MAM) and summer (JJA) (Yin et al., 2017) (top). The frequencies of the 48 h back trajectories of air masses arriving at Nam Co station from different directions during pre-monsoon and monsoon seasons in this study (bottom).

[Figure]

**Figure R3.** Comparison of meteorology between this study and the whole seasons in 2020. Time series of ambient temperature, wind speed and relative humidity at Nam Co station from January 2020 to December 2020 (National Tibetan Plateau Data Center) (left). Comparison in frequency distributions of temperature, WS and RH at Nam Co station in pre-monsoon and monsoon seasons in this study (right).

[Figure]

**Figure R4.** Comparison of atmospheric pollutants between this study and previous TP studies. Time series of f BC, aerosol scattering coefficient, $PM_1$, coarse particle number and surface ozone at the Nepal Climate Observatory-Pyramid (NCO-P) station from 1 March 2006 to 28 February 2008 (Bonasoni et al., 2010) (left). Comparison in frequency distributions of BC, $PM_{0.8}$ and $O_3$ at Nam Co station in pre-monsoon and monsoon seasons in this study (right).

3) Based on the above discussion, the two measurements periods in this study can respectively represent the pre-monsoon and monsoon seasons at Nam Co station, so the NPF characteristics of the two observation periods can also be considered as the NPF characteristics in pre-monsoon and monsoon seasons. In addition, the phenomenon of higher NPF frequency in monsoon season than pre-monsoon season was also found in the other sites in the Tibetan Plateau (TP). A 16-month measurements from 2006 to 2007 at Himalayan Nepal Climate Observatory at Pyramid (NCO-P) site on the southern TP showed NPF frequency of 38% in pre-monsoon season and 57% in monsoon season (Venzac et al., 2008). At Mt. Yulong on the southeastern TP, the NPF frequency was only 14% during pre-monsoon season (Shang et al., 2018). The NPF frequency of 15% in pre-monsoon season and 80% in monsoon season at Nam Co station was consistent with these studies, with more significant seasonal differences. The significant seasonal differences may be due to the fact that the occurrence of NPF is more sensitive to the monsoon in extremely clean background areas (such as Nam Co station and Mt. Yulong). In summary, our study emphasized the seasonal differences in NPF frequencies at Nam Co station, and the results was reliable.

Wang, B. and Fan, Z.: Choice of South Asian Summer Monsoon Indices, Bulletin of the American Meteorological Society, 80, 629-638, 10.1175/1520-0477(1999)080<0629:COSASM>2.0.CO;2, 1999.

Yin, X., Kang, S., de Foy, B., Cong, Z., Luo, J., Zhang, L., Ma, Y., Zhang, G., Rupakheti, D., and Zhang, Q.: Surface ozone at Nam Co in the inland Tibetan Plateau: variation, synthesis comparison and regional representativeness, Atmos. Chem. Phys., 17, 11293-11311, 10.5194/acp-17-11293-2017, 2017.

Cong, Z., Kang, S., Kawamura, K., Liu, B., Wan, X., Wang, Z., Gao, S., and Fu, P.: Carbonaceous aerosols on the south edge of the Tibetan Plateau: concentrations, seasonality and sources, Atmos. Chem. Phys., 15, 1573-1584, 10.5194/acp-15-1573-2015, 2015.

Bonasoni, P., Laj, P., Marinoni, A., Sprenger, M., Angelini, F., Arduini, J., Bonafè, U., Calzolari, F., Colombo, T., Decesari, S., Di Biagio, C., Di Sarra, A., Evangelisti, F., Duchi, R., Facchini, M. C., Fuzzi, S., Gobbi, G. P., Maione, M., Panday, A., Roccato, F., Sellegri, K., Venzac, H., Verza, G., Villani, P., Vuillermoz, E., and Cristofanelli, P.: Atmospheric Brown Clouds in the Himalayas: first two years of continuous observations at the Nepal Climate Observatory-Pyramid (5079 m), Atmospheric Chemistry and Physics, 10, 7515-7531, 10.5194/acp-10-7515-2010, 2010.

Xu, J., Zhang, Q., Shi, J., Ge, X., Xie, C., Wang, J., Kang, S., Zhang, R., and Wang, Y.: Chemical characteristics of submicron particles at the central Tibetan Plateau: insights from aerosol mass spectrometry, Atmos. Chem. Phys., 18, 427-443, 10.5194/acp-18-427-2018, 2018.

Junbo, W.: Daily meteorological Data of Nam Co Station China during 2019-2020, National Tibetan Plateau Data Center [dataset], 10.11888/Meteoro.tpdc.271782, 2021.

Venzac, H., Sellegri, K., Laj, P., Villani, P., Bonasoni, P., Marinoni, A., Cristofanelli, P., Calzolari, F., Fuzzi, S., Decesari, S., Facchini, M.-C., Vuillermoz, E., and Verza, G. P.: High frequency new particle formation in the Himalayas, Proceedings of the National Academy of Sciences, 105, 15666-15671, doi:10.1073/pnas.0801355105, 2008.

Shang, D., Hu, M., Zheng, J., Qin, Y., Du, Z., Li, M., Fang, J., Peng, J., Wu, Y., Lu, S., and Guo, S.: Particle number size distribution and new particle formation under the influence of biomass burning at a high altitude background site at Mt. Yulong (3410 m), China, Atmos. Chem. Phys., 18, 15687-15703, 10.5194/acp-18-15687-2018, 2018.

Due to harsh conditions and logistical limitations, our observation periods were limited. However, our conclusions are obvious and representative, and we will carry out more detailed observations for a longer period in the future if possible. To illustrate the representativeness of the observation periods, we have made supplements in the revised manuscript as follows:

"**2.1 Measurement site**

The measurement was conducted from 26 April to 22 May, 2019 and 15 June to 25 June, 2019, and can be representative of the pre-monsoon season and the summer monsoon season, respectively (Text S1) (Bonasoni et al., 2010; Cong et al., 2015)."

"**Text S1 The representativeness of the observation periods**

The measurement was conducted from 26 April to 22 May, 2019 and 15 June to 25 June, 2019, and can be representative of the pre-monsoon season and monsoon season, respectively.

Firstly, the intensity of Indian Summer Monsoon during the two measurements periods can represent that in the whole pre-monsoon and monsoon seasons, respectively. The intensity of Indian Summer Monsoon is an important indicator to distinguish the monsoon season. Here the intensity of Indian Summer Monsoon (ISM) was indicated by the ISM Index, which are defined by the negative outgoing longwave radiation anomalies (with respect to the climatological annual cycle) averaged over the Bay of Bengal–India region (10°–25°N, 70°–100°E) (Wang and Fan, 1999). As shown in Fig. S1, the measurement periods (green boxes) were in the pre-monsoon season (March-May) and monsoon season (June-September), respectively. And the IMS index during the two measurements periods were equivalent to those of the whole pre-monsoon season (average: -19.5 vs -20.7 W m$^{-2}$) and monsoon season (average: 27.0 vs 26.3 W m$^{-2}$), respectively.

Secondary, the characteristics of meteorology and atmospheric pollutants in the two measurements periods was generally in agreement with the previous long-term studies at Nam Co station and other sites in the Tibetan Plateau (TP) (Yin et al., 2017; Cong et al., 2015; Bonasoni et al., 2010; Xu et al., 2018). That is, the characteristics of meteorology (temperature, WS and RH) and atmospheric pollutants (PM, BC and ozone) in the two measurements periods were matched with those in the whole pre-monsoon and monsoon season at Nam Co station and other sites in the TP.

Therefore, the two observation periods are representative in the seasonal characteristics in pre-monsoon season and monsoon season, respectively.

[Figure]

**Figure S1.** The Indian Summer Monsoon (ISM) Index in 2019. The measurements periods are marked by the green boxes."

*2. Second, many of the conclusions made in the paper rely on SO2 and VOC concentrations. Unfortunately, there are very limited measurements on these 2 trace gases (only VOCs during the monsoon season), instead their concentrations were estimated from large-scale model simulations. The simulated concentrations may have large uncertainties, which are not quantified by any means in the paper.*

Thanks for the comment. The model evaluation including the meteorological fields and air pollutants has been added in the revised manuscript. In general, the results of model simulation of VOC and $SO_2$ show good performance in statistical parameters of model evaluation and correlation analysis with other tracers.

The modelled VOC and $SO_2$ could be used for the NPF analysis. The detailed information can be found in the revised manuscript as follow:

**"2.3 Model simulation**

[revised manuscript text omitted]

**"2.3 Text S2 Model simulation**

**Model Configurations**

The meteorological conditions were simulated using the Weather Research and Forecasting (WRF) (version 4.2.1) model with the FNL reanalysis dataset. The 6 h FNL data were obtained from the U.S. National Centre for Atmospheric Research (NCAR), with a spatial resolution of 1.0° × 1.0° (http://rda.ucar.edu/datasets/ds083.2/, last accessed on 28 April 2022). The physical parameterizations used in this study are the Thompson microphysical process, RRTMG longwave/shortwave radiation scheme; Noah land-surface scheme; MYJ boundary layer scheme; and modified Tiedtke cumulus parameterization scheme. The detailed configuration settings could be found in the works of Hu et al. (2016), Mao et al. (2022), Wang et al. (2021a).

The Community Multiscale Air Quality version 5.3.2 (CMAQv5.3.2) model, being one of the three-dimensional chemical transport models (CTMs) (Appel et al., 2021), configured with the gas-phase mechanism of SAPRC07tic and the aerosol module of AERO6i, was employed in this study to simulate the air quality over Tibet from 24 April to 24 May and 13 June to 27 June in 2019, which contains the observation period. Air quality simulations were performed with a horizontal resolution of 12 km. The corresponding domain covered Tibet and the surrounding countries and regions with 166 × 166 grids (Fig. S2), with the 18 layers in vertical resolution. The initial and boundary conditions were provided by the default profiles. The simulated results of the first two days were not included in the model analysis, which served as a spin-up and reduced the effects of the initial conditions on the simulated results.

**Model Evaluation**

Previous studies have investigated the impacts of meteorological conditions on the formation, transportation, and dissipation of air pollutants (Hu et al., 2016; Hua et al., 2021; Mao et al., 2022; Sulaymon et al., 2021b;

Sulaymon et al., 2021a). Therefore, the evaluation of the WRF model performance was carried out before the usage of its meteorological fields in the CMAQ simulations. The evaluation of the WRF model was achieved by comparing the predicted wind speed (WS, m/s), wind direction (WD, °) at 10 m above the surface, RH (%) and temperature (T, °C) to the observed values. Fig. S3 showed that WS was well simulated both in pre-monsoon and monsoon seasons. WD was well simulated in pre-monsoon season, and there seems to be some deviation in the simulation of north wind in monsoon season. The main reason about the deviation in WD may be due to the poor terrain and complicated weather conditions. Nevertheless, both simulations and measurements showed more frequent southerly winds during monsoon season. RH and temperature were well simulated in the whole periods (Fig. S4). The good model performance with the statistical metrics of WS, RH and temperature meeting the suggested benchmarks are shown in Table S1. Generally, the simulated meteorological fields were qualitied and can be further utilized in driving the CMAQ model

Fig. S5 showed the comparison of simulated hourly mean concentration about PM, O$_3$ and VOC in observation site, which were simulated by CMAQ. The statistical indices used in evaluating the CMAQ model were present in

Table S2. It can be seen that PM and O$_3$ meet the suggested benchmarks, which reflect the good model performance.

The observed VOC and predicted VOC in pre-monsoon season were compared to examine the model performance.

The benchmarks for VOC had not been reported, but the MFB (mean fractional bias) and MFE (mean fractional error) values are within the range reported in previous VOC modelling result (Hu et al., 2017). The correlation coefficient (R) between simulated and observed VOC is 0.41, which reflected that the model can fairly simulate the variation of VOC concentration. It should be noted that VOC was underpredicted on the whole, which may due to the uncertainty of the emission inventory.

[Figure]

                          **Figure S2.** WRF/CMAQ modeling domain

[Figure]

**Figure S3.** Comparison of simulated (in red dot-line) and observed (in blue dot) wind direction (WD, °) and wind speed (WS, m/s). Observed is 10 minutes mean data. Simulated is hourly mean data.

[Figure]

**Figure S4.** Comparison of simulated and observed RH (%) and temperature (T, °C). RH and temperature are hourly mean data.

[Figure]

**Figure S5.** Comparison of simulated and observed PM (µg/m³), O₃ (ppb) and VOC (ppb). PM, O₃ and VOC are hourly mean concentration.

**Table S1.** Model performance of meteorological factors at Nam Co station

|  | WS | | | | RH | | | | T | | | |
|---|---|---|---|---|---|---|---|---|---|---|---|---|
|  | MB | ME | RMSE | R | MB | ME | RMSE | R | MB | ME | RMSE | R |
| Statistic | 0.42 | 0.87 | 1.20 | 0.51 | -1.38 | 12.20 | 16.30 | 0.67 | 0.07 | 1.85 | 2.43 | 0.89 |
| Benchmarks | ≤±0.5 | ≤2.0 | ≤2.0 | | | | | | ≤±0.5 | ≤2.0 | | |

*MB: mean bias; ME: mean error; RMSE: root mean square error; R: correlation coefficient. The benchmarks were suggested by* Boylan and Russell (2006).

**Table S2.** Model performance of the air pollutants at Nam Co station

|  | PM₁ | | | O₃ | | | VOC | | | SO₂ | | |
|---|---|---|---|---|---|---|---|---|---|---|---|---|
|  | MFB | MFE | R | NMB | NME | R | MFB | MFE | R | NMB | NME | R |
| Statistic | 0.49 | 0.50 | 0.72 | 0.14 | 0.23 | 0.51 | -0.47 | 0.49 | 0.41 | | | |
| Benchmarks | <±0.6 | <0.75 | >0.4 | <±0.15 | <0.35 | >0.5 | | | | | | |
| References | | | | | | | <±0.77 | <0.74 | | <±4.38 | <±4.38 | 0.25-0.79 |

*NMB: normalized mean bias; NME: normalized mean error; R: correlation coefficient; MFB: mean fractional bias; MFE: mean fractional error. The benchmarks for PM and O₃ were suggested by Emery et al. (2017) and Boylan and Russell (2006), respectively. The references for VOC and SO₂ were from Hu et al. (2017) and Mao et al. (2022), respectively.*

[Figure]

**Figure S6.** Relationship between $SO_2$ and BC at Mt. Yulong in 2015. The correlation coefficient R is 0.79.

[Figure]

**Figure S7.** Relationship between modelled $SO_2$ and BC at Nam Co station. The correlation coefficient R is 0.58."

**Reply to Reviewer 1's important scientific comments (5):**

*1. The description on how CCN concentrations were calculated (section 2.4) is incomplete. Apparently, the authors used equations 4 and 5 to determine the critical diameters corresponding to different supersaturations, and from these critical diameters one then gets the number of CCN using measured particle number size distributions. However, this calculation cannot be done without knowing the hygroscopicity parameter cappa. Did the authors simply assumed a fixed value for cappa or did they estimate it from some chemical data?*

Thanks for the comment. The hygroscopicity parameter kappa was assumed to be a constant value of 0.12

throughout the measurement period, according to the previous measurement at Mt. Yulong on the southeastern TP (Shang et al., 2018). This is mainly due to the similar proportions of chemical components between Nam Co station (Xu et al., 2018) and Mt. Yulong with around 70% fraction of organics (Fig R5).

And the fractions of chemical components at Nam Co station were stable.

[Figure]

[Figure]

**Figure R5.** The combo plot of the data of the Nam Co study including (a) the meteorological conditions (T :

air temperature; RH: relative humidity; Precip.: precipitation), (b) the variation of WS (wind speed) colored according to WD (wind direction), (c) the temporal variation of mass concentration of $PM_1$ species and the average contribution of each species (pie chart), (d) the mass contribution of each $PM_1$ species and the total mass concentration of PM1, and (e) the mass contribution of PMF results. Three periods based on the meteorological conditions were marked (Xu et al., 2018) (top). The average contribution of each chemical species at Mt. Yulong on the southeastern TP (Shang et al., 2018) (bottom).

In addition, the kappa at Nam Co station in pre-monsoon and monsoon seasons can be estimated by using the previously measured chemical data at this site (Xu et al., 2018). The kappa of the mixed particles was calculated based on κ-Köhler theory and the Zdanovskii–Stokes–Robinson (ZSR) mixing rule (Petters and Kreidenweis, 2007). The values of κ is 0.48 for $(NH_4)_2SO_4$ and 0.58 for $NH_4NO_3$ (Petters and Kreidenweis, 2007). The κ is assumed to be 0.1 for organics (Wu et al., 2015). We derived the volume fraction of each species by dividing mass concentration by its density. The densities are $1.77\ g\,cm^{-3}$ for $(NH_4)_2SO_4$ and $1.72\ g\,cm^{-3}$ for $NH_4NO_3$. The densities of organics are assumed to be $1.2\ g\,cm^{-3}$ (Fan et al., 2020). The κ and density of BC are assumed to be 0 and $1.7\ g\,cm^{-3}$. It was found that average value of kappa was 0.15 and 0.13 in pre-monsoon and monsoon seasons, respectively. The $D_c$ at $S_c$ levels of 0.6% and 1.2% were 72.4±1.0 and 45.7±0.6 nm in pre-monsoon season, and 69.1±0.9 and 43.6±0.6 nm in monsoon season. It was comparable with that used in this study (κ: 0.12, $S_c$=0.6%: 73.4±1.3, $S_c$=1.2%: 46.3±0.8 nm). And it had little effect on the final result of CCN concentration. Therefore, we finally decided to adopt the fixed κ value of 0.12.

Shang, D., Hu, M., Zheng, J., Qin, Y., Du, Z., Li, M., Fang, J., Peng, J., Wu, Y., Lu, S., and Guo, S.: Particle number size distribution and new particle formation under the influence of biomass burning at a high altitude background site at Mt. Yulong (3410 m), China, Atmos. Chem. Phys., 18, 15687-15703, 10.5194/acp-18-15687-2018, 2018.

Xu, J., Zhang, Q., Shi, J., Ge, X., Xie, C., Wang, J., Kang, S., Zhang, R., and Wang, Y.: Chemical characteristics of submicron particles at the central Tibetan Plateau: insights from aerosol mass spectrometry, Atmos. Chem. Phys., 18, 427-443, 10.5194/acp-18-427-2018, 2018.

Wu, Z. J., Poulain, L., Birmili, W., Größ, J., Niedermeier, N., Wang, Z. B., Herrmann, H., and Wiedensohler, A.: Some insights into the condensing vapors driving new particle growth to CCN sizes on the basis of hygroscopicity measurements, Atmos. Chem. Phys., 15, 13071-13083, 10.5194/acp-15-13071-2015, 2015.

Petters, M. D. and Kreidenweis, S. M.: A single parameter representation of hygroscopic growth and cloud condensation nucleus activity, Atmos. Chem. Phys., 7, 1961-1971, 10.5194/acp-7-1961-2007, 2007.

Fan, X., Liu, J., Zhang, F., Chen, L., Collins, D., Xu, W., Jin, X., Ren, J., Wang, Y., Wu, H., Li, S., Sun, Y., and Li, Z.: Contrasting size-resolved hygroscopicity of fine particles derived by HTDMA and HR-ToF-AMS measurements between summer and winter in Beijing: the impacts of aerosol aging and local emissions, Atmos. Chem. Phys., 20, 915-929, 10.5194/acp-20-915-2020, 2020.

**"3.4 Significant increase of CCN in monsoon season**

The CCN concentration was estimated following the method introduced in Sect. 2.4. Considering the similar proportions of chemical components between Nam Co station and Mt. Yulong with around 70% fraction of organics, and the stability of the fractions of chemical components, the hygroscopicity parameter κ was assumed to be a constant value of 0.12 throughout the measurement period according to the previous measurement at Mt. Yulong in the TP (Shang et al., 2018)."

*2. The discussion on the role of condensation sink (CS) in favoring/disfavoring NPF is not logical. The authors first say that their result differs from those found in earlier studies (lines 243-244), but then mention a few studies which actually agree with their findings (lines 245-247). Please reformulate this part*

*of the text, as it causes confusion in its present form.*

Thanks for the comment. Considering the possible confusion, we have reformulated the discussion in the revised manuscript as follows:

"**3.3.1 Condensation sink**

 CS is the key factor controlling the occurrence of NPF events especially in urban environment (Yan et al.,

2021). At some high-altitude observations at a larger scale, the important role of the transported pre-existing particles in NPF events was also emphasized (Rose et al., 2019; Boulon et al., 2010). Here we analyzed the CS

levels in NPF days and non-event days at Nam Co station. As shown in Fig. 3a, the levels of CS in NPF-pre days,

NPF-monsoon days and non-event days were approximate during 11:00–18:00 (the occurrence time of NPF events), although the CS in the early morning of NPF-pre days seems to be slightly lower. The CS was mainly in the range of $0.1 \times 10^{-2}$-$0.15 \times 10^{-2}$ $s^{-1}$ during the NPF occurrence time, which was much lower than that at most locations in

China, such as ~$0.01$ $s^{-1}$ in urban Beijing (Deng et al., 2021), and $0.1 \times 10^{-2}$ to $28.4 \times 10^{-2}$ $s^{-1}$ at Mt. Tai (Lv et al.,

2018), and comparable with that at Mt. Yulong (~$0.2 \times 10^{-2}$ $s^{-1}$) (Shang et al., 2018). The result varied from previous studies which reported much lower CS during NPF days than that in non-event days (Zhou et al., 2021; Lv et al.,

2018). It indicated that CS was not the decisive factor controlling the occurrence of NPF events at Nam Co station."

**3. The discussion on the role of VOCs (lines 266-277) could be improved as well. First, considering the**

**typical variability of VOC concentrations in ambient measurements, I would think a 20% higher VOC**

**concentration is slightly rather than noticeably higher (line 268). It is also confusing that for the pre-**

**monsoon season the VOC concentration difference is given as % while for the monsoon season it is given**

**as an absolute value (ppb).**

Thanks for the comment. This part has been modified in the revised manuscript as follows:

"**3.3.1 Gas precursors**

 In addition to sulfuric acid, organics are also considered to be an important factor of NPF events. Observations and laboratory experiments have found that organics may participate in or even dominate the nucleation and growth process in NPF events in pristine environments and the preindustrial atmosphere. For example, CLOUD (Cosmics

Leaving Outdoor Droplets) experiments observed obvious NPF events from highly oxidized organics without the involvement of sulfuric acid (Kirkby et al., 2016). At the high-altitude sites of Jungfraujoch and Himalaya, NPF

events occurred mainly through the condensation of highly oxygenated molecules (HOMs) (Bianchi et al., 2016;

Bianchi et al., 2021). Due to instrument status, VOC measurement was only available in pre-monsoon season. The concentration of VOC (total VOC) showed a higher value (20%) during 11:00-18:00 on NPF-pre days compared with non-event days (Fig. 3c). Aromatics, which can be used as the indicator of anthropogenic emissions, also exhibited a higher level (20%) during NPF-pre days (Fig. 3d). This suggested that VOC may be the key factor in determining the occurrence of NPF events. In order to further evaluate the role of VOC, we used WRF/CMAQ models to simulate the spatial distribution of VOC concentration in pre-monsoon and monsoon seasons. The detailed information about the model setting and evaluation can be found in Text S2. As shown in Fig.4, the simulated VOC levels in NPF days including NPF-pre and NPF-monsoon days were higher than those in non-event days. The average modelled VOC concentrations in NPF-pre days and NPF-monsoon days were 25% and 88% higher than those in non-event days, respectively. Therefore, we further considered that the organic matter could be the key factor to determine whether NPF event occurred. Nucleation of pure organics or organics involved could be the dominant mechanism at Nam Co station. In addition, higher organic concentrations were observed in monsoon season (NPF-monsoon days) compared with those in pre-monsoon season (NPF-pre days and non-event days). The result is consistent with one recent research which has found that the concentration of monoterpene-derived HOMs in East Asia was higher in summer (June-August) than that in Spring (March-May) by using GEOS-Chem global chemical transport model (Xu et al., 2022). This means that the frequent NPF events in monsoon season could be caused by the higher organic matters.

**4. Wind direction is a very local quantity, and does not necessary tell correctly air pollutant sources or transport pathways. I wonder whether the authors have information on air mass trajectories which would provide more direct support for their statements on lines 283-295.**

Thanks for the comment. The air mass trajectories have been added in the revised manuscript as follpws:

"**3.3.1 Air mass origins and meteorology**

While the concentrations of organic precursors could be the most possible reason for the occurrence of NPF events, the external factors driving the difference in VOC levels between the NPF and non-event days and other conditions that may affect the characteristics of NPF were still unknown. This indicated that a further investigation into other NPF-related variables was still required.

There are almost no local anthropogenic source emissions at Nam Co station. The air pollutants at the observation site mainly brought by air mass transmission. Backward trajectories were utilized to identify the air mass origins associated with NPF events. The frequencies of the 48 h back trajectories of air masses arriving at Nam Co station during the occurrence time of nucleation (11:00-18:00) in non-event days, NPF-pre days and NPF-monsoon days were present in Fig. 5. It can be found that the dominant air masses in non-event days were from the west (almost 100%) and passed by western Nepal, northwest India and Pakistan. In comparison, air masses in NPF-pre days and NPF-monsoon days mainly come from the south and southwest (57% and 75%) and originated in the northeast India. WD, which can reflect the local situation for air masses and the source of air pollutants, showed a high frequency of strong southerly wind in NPF-pre days and NPF-monsoon days and westerly wind in non-event days (Fig. S14). The example of the spatial distribution of wind field in non-event days, NPF-pre days and NPF- monsoon days displayed the similar phenomenon on a larger scale (Fig. S15). These results suggested that the occurrence of NPF events was related to the southerly and southwesterly air masses. When the southerly air mass occurred, it may bring organic precursors from the southern region (northeast India) to Nam Co station, driving the occurrence of NPF events in this area. As for the significantly higher NPF frequency in monsoon season, it resulted from the more frequent southerly air masses (summer monsoon) in monsoon season in comparison with pre-monsoon season as introduced in Sec. 3.1. The summer monsoon can bring the higher organic concentrations in monsoon season (NPF-monsoon days) compared with those in pre-monsoon season (NPF-pre and non-event days) (Fig.4), thus triggered almost daily NPF events. Similar result was found in the recent study which showed that the Indian summer monsoon acted as a facilitator for transporting gaseous pollutants (Yin et al., 2021).

[Figure]

**Figure 4.** The frequencies of the 48 h back trajectories of air masses arriving at Nam Co station from different directions during the occurrence time of nucleation (11:00-18:00) in (a) non-event days, (b) NPF-pre days and (c) NPF-monsoon days."

***5. The enhancements of CCN concentrations due to NPF is reported in 3 different ways in section 3.4: 1) using an enhancement factor EF, 2) using percentage increases, and 3) stating that something is N times higher than... This is confusing. I highly recommend the authors to unify this discussion.***

Thanks for the comment. Considering the possible confusion, we have unified the discussion (N times higher than) in the revised manuscript as follows:

"**3.4 Significant increase of CCN in monsoon season**

[revised manuscript text omitted]

**Reply to Reviewer 1's minor comments (5):**

*1. line 49: I suppose that the authors mean quantities like the particle formation and growth rate as referring to parameters of NPF. I do not feel that parameter is a good wording here, rather suggesting something as characteristics of NPF.*

Thanks for the help comment. We have made correction in the revised manuscript.

*2. lines 127-128: Classification of a NPF event seems untypical. Has the performance of this classification method tested and has it been used in other studies besides Fang et al. (2020)? The word obviously does not fit into this context.*

Thanks for the comment. The classification method of NPF events has been used in Tang et al (2021) and Shang et al (2022). Considering the possible misunderstanding, we have described the classification method in more detail in the revised manuscript.

Tang, L., Shang, D., Fang, X., Wu, Z., Qiu, Y., Chen, S., Li, X., Zeng, L., Guo, S., and Hu, M.: More Significant Impacts From New Particle Formation on Haze Formation During COVID-19 Lockdown, Geophysical Research Letters, 48, e2020GL091591, https://doi.org/10.1029/2020GL091591, 2021.

Shang, D., Tang, L., Fang, X., Wang, L., Yang, S., Wu, Z., Chen, S., Li, X., Zeng, L., Guo, S., and Hu, M.: Variations in source contributions of particle number concentration under long-term emission control in winter of urban Beijing, Environmental Pollution, 304, 119072, https://doi.org/10.1016/j.envpol.2022.119072, 2022.

"**2.3 Parameterization of NPF**

In this study, a typical NPF event was defined by that there is a burst in the 3-10 nm particles number concentration ($PN_{3-10}$) and subsequent growth of these newly formed particles (Fang et al., 2020; Dal Maso et al., 2005). The days without newly particle formation were defined as non-event days. Days in which the increase of $PN_{3-10}$ was observed without the particles growing to the larger size, or days when we can see the later phase of a mode growing in the Aitken mode size range were treated as undefined days (Dal Maso et al., 2005). The examples of the classification of NPF events are shown in Fig.S8.

Fang, X., Hu, M., Shang, D., Tang, R., Shi, L., Olenius, T., Wang, Y., Wang, H., Zhang, Z., Chen, S., Yu, X., Zhu, W., Lou, S., Ma, Y., Li, X., Zeng, L., Wu, Z., Zheng, J., and Guo, S.: Observational Evidence for the Involvement of Dicarboxylic Acids in Particle Nucleation, Environmental Science & Technology Letters, 7, 388-394, 10.1021/acs.estlett.0c00270, 2020.

Dal Maso, M., Kulmala, M., Riipinen, I., and Wagner, R.: Formation and growth of fresh atmospheric aerosols: Eight years of aerosol size distribution data from SMEAR II, Hyytiälä, Finland, Boreal Environment Research, 10, 323-336, 2005.

[Figure]

**Figure S8.** Typical particle number size distributions of (a) a NPF day (23 June, 2019), (b) an undefined day (24 June, 2019), and (c) a non-event day (15 May, 2019)."

*3. lines 225-233: The authors list a number of things that potentially affect the occurrence of NPF. The list misses one highly relevant quantity: the intensity of solar radiation. This quantity should be mentioned here.*

Thanks for the comment. We have made correction in the revised manuscript as follows:

"Whether an NPF event can occur is mainly related to 1) the CS, which mainly referred to the scavenging rate of precursors, clusters, and newly formed particles by background aerosols. High CS can lead to the continual reduction in newly formed particle number concentration, and inhibit the occurrence of NPF; 2) the gaseous precursors that can participate in nucleation and growth, including sulfuric acid (Kulmala et al., 2013), dimethylamine (Yao et al., 2018), ammonia (Xiao et al., 2015) and VOC (Tröstl et al., 2016; Fang et al., 2020; Qiao et al., 2021). A sufficiently high concentration of low volatility vapors (precursors) can contribute to persistent nucleation and generating new atmospheric particles; 3) air mass origins and meteorological factors including WD, RH, temperature, the intensity of solar radiation, etc, which can influence the occurrence and intensity of NPF events by directly or indirectly affecting the source and sink parameters."

*4. Unlike the Aitken mode, the nucleation mode is usually written using a lower-case letter (lines 326, 359, 383)*

Thanks for the comment. The nucleation mode is written using a lower-case letter in the revised manuscript:

"nucleation mode particles"

*5. In several places (lines 225, 254, 258, 261, 284, 296, 299, 340), the use of tense is somewhat wrong, or at least uncommon. Please reconsider which tense to use in these places.*

Thanks for the comment. We have made correction in the revised manuscript as follows:

"Whether an NPF event can occur is mainly related to 1) the CS, which mainly referred to the scavenging rate of precursors, clusters, and newly formed particles by background aerosols."

"Gaseous sulfuric acid is identified as the key precursor for nucleation and initial growth due to its low volatility (Kulmala et al., 2013; Qiao et al., 2021)."

"In addition to sulfuric acid, organics are also considered to be an important factor of NPF events. Observations and laboratory experiments have found that organics may participate in or even dominate the nucleation and growth process in NPF events in pristine environments and the preindustrial atmosphere."

"The frequencies of the 48 h back trajectories of air masses arriving at Nam Co station during the occurrence time of nucleation (11:00-18:00) in non-event days, NPF-pre days and NPF-monsoon days are present in Fig. 5."

"In Fig. 6, we show the diurnal variations of meteorological factors during NPF-pre days, NPF-monsoon days and non-event days at Nam Co station."

"The similar temperature in NPF-pre days and non-event days suggests that temperature is not a crucial factor for NPF event occurrence."

"The average PNSD during pre-monsoon and monsoon seasons are plotted in Fig. 7a with much higher number concentrations observed during monsoon season."

---

## Author Comment (AC2)

**Reply to comments**

Journal: Atmospheric Chemistry and Physics

Manuscript Number: acp-2022-440

Title: "High frequency of new particle formation events driven by summer monsoon in the central Tibetan Plateau, China"

Author(s): Lizi Tang, Min Hu, Dongjie Shang, Xin Fang, Jianjiong Mao, Wanyun Xu, Jiacheng Zhou, Weixiong Zhao, Yaru Wang, Chong Zhang, Yingjie Zhang, Jianlin Hu, Limin Zeng, Chunxiang Ye, Song Guo, Zhijun Wu

**I. Reply to Reviewer 2**

**Reply to Reviewer 2's overall comments:**

*New particle formation (NPF) at high altitudes is crucial to understand sources of aerosol and CCN in the free troposphere. In this study, the authors conducted intensive measurements at Nam Co station (4379 m a.s.l) in the central TP to understand the new particle formation during pre-monsoon and monsoon seasons. They identified the frequency of NPF during monsoon seasons was significantly higher than during pre-monsoon seasons. This study did provide valuable observation data. But the explanation that higher VOCs triggered the frequent NPF during monsoon season is unconvincing. Therefore, the manuscript is not recommended to be published on ACP unless the authors can address the following major concerns.*

> We appreciate the comments from the reviewer on this manuscript. We have answered them point to point in the following paragraphs (the texts italicized are the comments, the texts indented are the responses, and the texts in blue are revised parts in new manuscript). In addition, all changes made are marked in the revised manuscript.

**Reply to Reviewer 2's comments (3):**

**1. Sulfuric acid (SA), Ultra/Extremely Low Volatility Organic Compounds (U/ELVOCs), and bases, e.g., NH3 or DMA, are known as the essential precursor of NPF. Their concentrations determine whether NPF can occur, as well as the intensity. However, in this study, all these key precursors were not measured. Even the precursors of these "direct precursors", e.g. SO2 and VOCs who can form low-volatile oxidation products, were not well measured either. First, the simulated concentration of SO2 used in this study without any verification by observation data is not convincing. Since SO2 is a very reactive species, one**

*needs to use the simulated value very carefully. Second, although 99 types of VOCs were measured during pre-monsoon using a GC-MS/FID, they are key precursors of ozone formation and are not suitable as indicator precursors for ELVOCs. The author, at least, needs to provide the concentration of monoterpenes, which are well-known sources of ELVOCs. In addition, the simulation of VOCs during monsoon is needed to be verified. In summary, the authors need to provide more solid evidence to support their main conclusion that higher VOCs triggered the frequent NPF during monsoon season.*

Thanks for the comment. The measurements of the NPF precursors including $SO_2$ and VOC are limited due to the harsh conditions and logistical limitations, with only VOC in pre-monsoon season. So we utilized WRF/CMAQ modeling system to simulate the levels of $SO_2$ and VOC in the whole observation period, to assist in the analysis of the role of sulfuric acid and organics. As for the verification of the simulated $SO_2$ and VOC, we have added the analysis on statistical parameters of model evaluation and correlation analysis with other tracers in the revised manuscript. Firstly, the WRF/CMAQ models successfully reproduced the meteorological fields and air pollutants including PM and $O_3$ with model performance indices meeting the suggested benchmarks, which means the simulated meteorological fields and particulate and gaseous pollutants are qualited. For VOC, the observed VOC and predicted VOC in pre-monsoon season were compared to examine the model performance. The benchmarks for VOC had not been reported, but the statistical metrics of MFB (mean fractional bias, -0.47) and MFE (mean fractional error, 0.49) in this study are within the range reported in previous VOC modelling result (Hu et al., 2017). The correlation coefficient (R) between simulated and observed VOC is 0.41, which reflected that the model can fairly simulate the variation of VOC concentration. For $SO_2$, the WRF/CMAQ models have been successfully reproduced $SO_2$ in major regions in China with R of 0.25-0.79 (Mao et al., 2022). The simulated $SO_2$ level in the model domain is comparable with that measured at Mt. Yulong (Shang et al., 2018), with average values of $0.03\pm0.02$ ppbv and $0.06\pm0.05$ ppbv. At the same time, considering that both BC and $SO_2$ are mainly emitted from coal combustion and biomass burning, BC could be a good indicator for $SO_2$ especially for pristine environment without local anthropogenic source emissions. A good correlation between $SO_2$ and BC measured at Mt. Yulong was found with correlation coefficient (R) of 0.79 (Shang et al., 2018). In this study, the modelled $SO_2$ and measured BC also showed good correlation with R of 0.58. In general, the results of model simulation showed good performance in statistical parameters and correlation analysis with other tracers. The modelled VOC and $SO_2$ could be used for the NPF analysis.

As for monoterpene such as α-pinene, unfortunately, it was not measured in this study. And there can be some other reference compound such as OVOC for understanding new particle formation from tree

emissions as indicated by the plant chamber experiment (Mentel et al., 2009). In addition, recent studies showed that aromatic compounds such as benzene, toluene, and naphthalene, and C6–C10 alkanes can produce considerable amounts of highly oxygenated products through multi-generation OH oxidation or autoxidation (Garmash et al., 2020; Wang et al., 2021), which may trigger the occurrence of NPF events. Therefore, we prefer that different VOC can affect the occurrence of NPF, and we mainly use the concentration of total VOC for analysis.

On the whole, we examined the potential reasons for the distinct NPF frequency using the measured CS, precursors, meteorology and simulated $SO_2$ and VOC. The comprehensive analysis points to the important role of organics. The higher NPF frequency driven by the higher organics concentration in monsoon season can be supported in one recent research which has found that the concentration of monoterpene-derived HOMs in East Asia was higher in summer (June-August) than that in Spring (March-May) by using GEOS-Chem global chemical transport model (Xu et al., 2022).

Hu, J., Chen, J., Ying, Q., and Zhang, H.: One-year simulation of ozone and particulate matter in China using WRF/CMAQ modeling system, Atmos. Chem. Phys., 16, 10333-10350, 10.5194/acp-16-10333-2016, 2016.

Shang, D., Hu, M., Zheng, J., Qin, Y., Du, Z., Li, M., Fang, J., Peng, J., Wu, Y., Lu, S., and Guo, S.: Particle number size distribution and new particle formation under the influence of biomass burning at a high altitude background site at Mt. Yulong (3410 m), China, Atmos. Chem. Phys., 18, 15687-15703, 10.5194/acp-18-15687-2018, 2018.

Mao, J., Li, L., Li, J., Sulaymon, I. D., Xiong, K., Wang, K., Zhu, J., Chen, G., Ye, F., Zhang, N., Qin, Y., Qin, M., and Hu, J.: Evaluation of Long-Term Modeling Fine Particulate Matter and Ozone in China During 2013–2019, Frontiers in Environmental Science, 10, 10.3389/fenvs.2022.872249, 2022.

Mentel, T. F., Wildt, J., Kiendler-Scharr, A., Kleist, E., Tillmann, R., Dal Maso, M., Fisseha, R., Hohaus, T., Spahn, H., Uerlings, R., Wegener, R., Griffiths, P. T., Dinar, E., Rudich, Y., and Wahner, A.: Photochemical production of aerosols from real plant emissions, Atmos. Chem. Phys., 9, 4387-4406, 10.5194/acp-9-4387-2009, 2009.

Garmash, O., Rissanen, M. P., Pullinen, I., Schmitt, S., Kausiala, O., Tillmann, R., Zhao, D., Percival, C., Bannan, T. J., Priestley, M., Hallquist, Å. M., Kleist, E., Kiendler-Scharr, A., Hallquist, M., Berndt, T., McFiggans, G., Wildt, J., Mentel, T. F., and Ehn, M.: Multi-generation OH oxidation as a source for highly oxygenated organic molecules from aromatics, Atmos. Chem. Phys., 20, 515-537, 10.5194/acp-20-515-2020, 2020.

Wang, Z., Ehn, M., Rissanen, M. P., Garmash, O., Quéléver, L., Xing, L., Monge-Palacios, M., Rantala, P., Donahue, N. M., Berndt, T., and Sarathy, S. M.: Efficient alkane oxidation under combustion engine and atmospheric conditions, Communications Chemistry, 4, 18, 10.1038/s42004-020-00445-3, 2021.

[revised manuscript text omitted]

**"2.3 Text S2 Model simulation**

**Model Configurations**

The meteorological conditions were simulated using the Weather Research and Forecasting (WRF) (version 4.2.1) model with the FNL reanalysis dataset. The 6 h FNL data were obtained from the U.S. National Centre for Atmospheric Research (NCAR), with a spatial resolution of 1.0° × 1.0° (http://rda.ucar.edu/datasets/ds083.2/, last accessed on 28 April 2022). The physical parameterizations used in this study are the Thompson microphysical process, RRTMG longwave/shortwave radiation scheme; Noah land-surface scheme; MYJ boundary layer scheme; and modified Tiedtke cumulus parameterization scheme. The detailed configuration settings could be found in the works of Hu et al. (2016), Mao et al. (2022), Wang et al. (2021a).

The Community Multiscale Air Quality version 5.3.2 (CMAQv5.3.2) model, being one of the three-dimensional chemical transport models (CTMs) (Appel et al., 2021), configured with the gas-phase mechanism of SAPRC07tic and the aerosol module of AERO6i, was employed in this study to simulate the air quality over Tibet from 24 April to 24 May and 13 June to 27 June in 2019, which contains the observation period. Air quality simulations were performed with a horizontal resolution of 12 km. The corresponding domain covered Tibet and the surrounding countries and regions with 166 × 166 grids (Fig. S2), with the 18 layers in vertical resolution. The initial and boundary conditions were provided by the default profiles. The simulated results of the first two days were not included in the model analysis, which served as a spin-up and reduced the effects of the initial conditions on the simulated results.

**Model Evaluation**

Previous studies have investigated the impacts of meteorological conditions on the formation, transportation, and dissipation of air pollutants (Hu et al., 2016; Hua et al., 2021; Mao et al., 2022; Sulaymon et al., 2021b; Sulaymon et al., 2021a). Therefore, the evaluation of the WRF model performance was carried out before the usage of its meteorological fields in the CMAQ simulations. The evaluation of the WRF model was achieved by comparing the predicted wind speed (WS, m/s), wind direction (WD, °) at 10 m above the surface, RH (%) and temperature (T, ℃) to the observed values. Fig. S3 showed that WS was well simulated both in pre-monsoon and monsoon seasons. WD was well simulated in pre-monsoon season, and there seems to be some deviation in the simulation of north wind in monsoon season. The main reason about the deviation in WD may be due to the poor terrain and complicated weather conditions. Nevertheless, both simulations and measurements showed more frequent southerly winds during monsoon season. RH and temperature were well simulated in the whole periods (Fig. S4). The good model performance with the statistical metrics of WS, RH and temperature meeting the

suggested benchmarks are shown in Table S1. Generally, the simulated meteorological fields were qualitied and can be further utilized in driving the CMAQ model

Fig. S5 showed the comparison of simulated hourly mean concentration about PM, $O_3$ and VOC in observation site, which were simulated by CMAQ. The statistical indices used in evaluating the CMAQ model were present in Table S2. It can be seen that PM and $O_3$ meet the suggested benchmarks, which reflect the good model performance. The observed VOC and predicted VOC in pre-monsoon season were compared to examine the model performance. The benchmarks for VOC had not been reported, but the MFB (mean fractional bias) and MFE (mean fractional error) values are within the range reported in previous VOC modelling result (Hu et al., 2017). The correlation coefficient (R) between simulated and observed VOC is 0.41, which reflected that the model can fairly simulate the variation of VOC concentration. It should be noted that VOC was underpredicted on the whole, which may due to the uncertainty of the emission inventory.

[Figure]

**Figure S2.** WRF/CMAQ modeling domain

[Figure]

**Figure S3.** Comparison of simulated (in red dot-line) and observed (in blue dot) wind direction (WD, °) and wind speed (WS, m/s). Observed is 10 minutes mean data. Simulated is hourly mean data.

[Figure]

**Figure S4.** Comparison of simulated and observed RH (%) and temperature (T, °C). RH and temperature are hourly mean data.

[Figure]

**Figure S5.** Comparison of simulated and observed PM (µg/m³), O₃ (ppb) and VOC (ppb). PM, O₃ and VOC are hourly mean concentration.

**Table S1.** Model performance of meteorological factors at Nam Co station

|  | WS | | | | RH | | | | T | | | |
|---|---|---|---|---|---|---|---|---|---|---|---|---|
|  | MB | ME | RMSE | R | MB | ME | RMSE | R | MB | ME | RMSE | R |
| **Statistic** | 0.42 | 0.87 | 1.20 | 0.51 | -1.38 | 12.20 | 16.30 | 0.67 | 0.07 | 1.85 | 2.43 | 0.89 |
| **Benchmarks** | ≤±0.5 | ≤2.0 | ≤2.0 | | | | | | ≤±0.5 | ≤2.0 | | |

*MB: mean bias; ME: mean error; RMSE: root mean square error; R: correlation coefficient. The benchmarks were suggested by Boylan and Russell (2006).*

**Table S2.** Model performance of the air pollutants at Nam Co station

|  | PM₁ | | | O₃ | | | VOC | | | SO₂ | | |
|---|---|---|---|---|---|---|---|---|---|---|---|---|
|  | MFB | MFE | R | NMB | NME | R | MFB | MFE | R | NMB | NME | R |
| **Statistic** | 0.49 | 0.50 | 0.72 | 0.14 | 0.23 | 0.51 | -0.47 | 0.49 | 0.41 | | | |
| **Benchmarks** | <±0.6 | <0.75 | >0.4 | <±0.15 | <0.35 | >0.5 | | | | | | |
| **References** | | | | | | | <±0.77 | <0.74 | | <±4.38 | <±4.38 | 0.25-0.79 |

*NMB: normalized mean bias; NME: normalized mean error; R: correlation coefficient; MFB: mean fractional bias; MFE: mean fractional error. The benchmarks for PM and O₃ were suggested by Emery et al. (2017) and Boylan and Russell (2006), respectively. The references for VOC and SO₂ were from Hu et al. (2017) and Mao et al. (2022), respectively.*

[Figure]

**Figure S6.** Relationship between $SO_2$ and BC at Mt. Yulong in 2015. The correlation coefficient R is 0.79.

[Figure]

**Figure S7.** Relationship between modelled $SO_2$ and BC at Nam Co station. The correlation coefficient R is 0.58."

*2. The observation period is a bit too short, especially, with only 10 days during monsoon. One cannot be sure that the high NPF frequency observed during this 10-day observation can be representative of the entire monsoon period.*

Thanks for the comment. The measurements periods were a little short as the reviewer described, with about 4 weeks for the pre-monsoon season and 10 days for the monsoon season. But our measurements periods can be representative for this location during pre-monsoon season and monsoon season as follows: 1) The intensity of Indian Summer Monsoon during the two measurements periods can represent that in the whole pre-monsoon and monsoon seasons, respectively. The intensity of Indian Summer Monsoon is

an important indicator to distinguish the monsoon season. Here the intensity of Indian Summer Monsoon (ISM) was indicated by the ISM Index, which are defined by the negative outgoing longwave radiation anomalies (with respect to the climatological annual cycle) averaged over the Bay of Bengal–India region (10°–25°N, 70°–100°E) (Wang and Fan, 1999). As shown in Fig. R1, the measurement periods (green boxes) were in the pre-monsoon season (March-May) and monsoon season (June-September), respectively. And the IMS index during the two measurements periods were equivalent to those of the whole pre-monsoon season (average: -19.5 vs -20.7 W m$^{-2}$) and monsoon season (average: 27.0 vs 26.3 W m$^{-2}$), respectively. Therefore, we considered that these two observation periods are representative in the seasonal characteristics in pre-monsoon season and monsoon season, respectively.

[Figure]

**Figure R1.** The Indian Summer Monsoon (ISM) Index in 2019. The measurements periods are marked by the green boxes.

2) The characteristics of meteorology and atmospheric pollutants in the two measurements periods was generally in agreement with the previous long-term studies at Nam Co station and other sites in the Tibetan Plateau (TP) (Yin et al., 2017; Cong et al., 2015; Bonasoni et al., 2010; Xu et al., 2018). Both previous research and this study showed that, strong westerlies pass through western Nepal, northwest India and Pakistan in pre-monsoon season, while air masses were mainly derived from Bangladesh and northeast India and brought moisture that originated in the Bay of Bengal in monsoon season (Fig. R2) (Yin et al., 2017). The temperature, WS and RH in the two measurements periods were matched with those in the whole pre-monsoon and monsoon season of 2020 at Nam Co station (Fig. R3) (National Tibetan Plateau Data Center). The average temperature in pre-monsoon and monsoon seasons were around 3 and 10 ℃, respectively. WS showed no difference between pre-monsoon and monsoon seasons with the average value of 4 m/s. The average level of RH was similar between the two seasons, but the variation range of

RH was larger in pre-monsoon season. In addition, the level of PM, BC and ozone in the two measurements periods were matched with those in the whole pre-monsoon and monsoon season at the other site of the TP (Fig. R4) (Bonasoni et al., 2010; Xu et al., 2018). The average concentrations of $PM_1$ ($PM_{0.8}$) in pre-monsoon and monsoon seasons were around 1 and 2 $\mu g/m^3$, respectively. The concentration of BC was at a level of hundreds nanogram per cubic meter in pre-monsoon season, while at a lower level in monsoon season. Ozone showed the similar pattern with BC. It should be noted that there will be some differences between this study and other results due to the differences in time resolution. In a word, the characteristics of meteorology and atmospheric pollutants in the two measurements periods in this study can well reflect those in pre-monsoon and monsoon seasons.

[Figure]

**Figure R2**. Comparison of trajectories between this study and previous study at Nam Co station. Backward HYSPLIT trajectories for each measurement day (black lines in the maps), and mean back trajectory for six HYSPLIT clusters (colored lines in the maps) arriving at Nam Co Station in spring (MAM) and summer (JJA) (Yin et al., 2017) (top). The frequencies of the 48 h back trajectories of air masses arriving at Nam Co station from different directions during pre-monsoon and monsoon seasons in this study (bottom).

[Figure]

**Figure R3.** Comparison of meteorology between this study and the whole seasons in 2020. Time series of ambient temperature, wind speed and relative humidity at Nam Co station from January 2020 to December 2020 (National Tibetan Plateau Data Center) (left). Comparison in frequency distributions of temperature, WS and RH at Nam Co station in pre-monsoon and monsoon seasons in this study (right).

[Figure]

**Figure R4.** Comparison of atmospheric pollutants between this study and previous TP studies. Time series of f BC, aerosol scattering coefficient, PM₁, coarse particle number and surface ozone at the Nepal Climate Observatory-Pyramid (NCO-P) station from 1 March 2006 to 28 February 2008 (Bonasoni et al., 2010) (left). Comparison in frequency distributions of BC, $PM_{0.8}$ and $O_3$ at Nam Co station in pre-monsoon and monsoon seasons in this study (right).

3) Based on the above discussion, the two measurements periods in this study can respectively represent the pre-monsoon and monsoon season at Nam Co station, so the NPF characteristics of the two observation periods can also be considered as the NPF characteristics in pre-monsoon and monsoon seasons. It is true that we can not be sure that the extremely high NPF frequency observed during the 10-day observation period can be found in the entire monsoon season. But the difference of NPF frequency between the two seasons is very notable. Here we do not emphasize the absolute value of NPF frequency, but the significant difference of NPF frequency between two seasons. In addition, the phenomenon of higher NPF frequency in monsoon season than pre-monsoon season was also found in the other sites in the Tibetan Plateau (TP). A 16-month measurements from 2006 to 2007 at Himalayan Nepal Climate

Observatory at Pyramid (NCO-P) site on the southern TP showed NPF frequency of 38% in pre-monsoon season and 57% in monsoon season (Venzac et al., 2008). At Mt. Yulong on the southeastern TP, the NPF frequency was only 14% during pre-monsoon season (Shang et al., 2018). The NPF frequency of 15% in pre-monsoon season and 80% in monsoon season at Nam Co station was consistent with these studies, with more significant seasonal differences. The significant seasonal differences may be due to the fact that the occurrence of NPF is more sensitive to the monsoon in extremely clean background areas (such as Nam Co station and Mt. Yulong). In summary, our study emphasized the seasonal differences in NPF frequencies at Nam Co station, and the results was reliable.

Wang, B. and Fan, Z.: Choice of South Asian Summer Monsoon Indices, Bulletin of the American Meteorological Society, 80, 629-638, 10.1175/1520-0477(1999)080<0629:COSASM>2.0.CO;2, 1999.

Yin, X., Kang, S., de Foy, B., Cong, Z., Luo, J., Zhang, L., Ma, Y., Zhang, G., Rupakheti, D., and Zhang, Q.: Surface ozone at Nam Co in the inland Tibetan Plateau: variation, synthesis comparison and regional representativeness, Atmos. Chem. Phys., 17, 11293-11311, 10.5194/acp-17-11293-2017, 2017.

Cong, Z., Kang, S., Kawamura, K., Liu, B., Wan, X., Wang, Z., Gao, S., and Fu, P.: Carbonaceous aerosols on the south edge of the Tibetan Plateau: concentrations, seasonality and sources, Atmos. Chem. Phys., 15, 1573-1584, 10.5194/acp-15-1573-2015, 2015.

Bonasoni, P., Laj, P., Marinoni, A., Sprenger, M., Angelini, F., Arduini, J., Bonafè, U., Calzolari, F., Colombo, T., Decesari, S., Di Biagio, C., Di Sarra, A., Evangelisti, F., Duchi, R., Facchini, M. C., Fuzzi, S., Gobbi, G. P., Maione, M., Panday, A., Roccato, F., Sellegri, K., Venzac, H., Verza, G., Villani, P., Vuillermoz, E., and Cristofanelli, P.: Atmospheric Brown Clouds in the Himalayas: first two years of continuous observations at the Nepal Climate Observatory-Pyramid (5079 m), Atmospheric Chemistry and Physics, 10, 7515-7531, 10.5194/acp-10-7515-2010, 2010.

Xu, J., Zhang, Q., Shi, J., Ge, X., Xie, C., Wang, J., Kang, S., Zhang, R., and Wang, Y.: Chemical characteristics of submicron particles at the central Tibetan Plateau: insights from aerosol mass spectrometry, Atmos. Chem. Phys., 18, 427-443, 10.5194/acp-18-427-2018, 2018.

Junbo, W.: Daily meteorological Data of Nam Co Station China during 2019-2020, National Tibetan Plateau Data Center [dataset], 10.11888/Meteoro.tpdc.271782, 2021.

Venzac, H., Sellegri, K., Laj, P., Villani, P., Bonasoni, P., Marinoni, A., Cristofanelli, P., Calzolari, F., Fuzzi, S., Decesari, S., Facchini, M.-C., Vuillermoz, E., and Verza, G. P.: High frequency new particle formation in the Himalayas, Proceedings of the National Academy of Sciences, 105, 15666-15671, doi:10.1073/pnas.0801355105, 2008.

Shang, D., Hu, M., Zheng, J., Qin, Y., Du, Z., Li, M., Fang, J., Peng, J., Wu, Y., Lu, S., and Guo, S.: Particle number size distribution and new particle formation under the influence of biomass burning at a high altitude background site at Mt. Yulong (3410 m), China, Atmos. Chem. Phys., 18, 15687-15703, 10.5194/acp-18-15687-2018, 2018.

Due to harsh conditions and logistical limitations, our observation periods were limited. However, our conclusions are obvious and representative, and we will carry out more detailed observations for a longer period in the future if possible. To illustrate the representativeness of the observation periods, we have made supplements in the revised manuscript as follows:

**"2.1 Measurement site**

The measurement was conducted from 26 April to 22 May, 2019 and 15 June to 25 June, 2019, and can be representative of the pre-monsoon season and the summer monsoon season, respectively (Text S1) (Bonasoni et al., 2010; Cong et al., 2015)."

"**Text S1 The representativeness of the observation periods**

The measurement was conducted from 26 April to 22 May, 2019 and 15 June to 25 June, 2019, and can be representative of the pre-monsoon season and monsoon season, respectively.

Firstly, the intensity of Indian Summer Monsoon during the two measurements periods can represent that in the whole pre-monsoon and monsoon seasons, respectively. The intensity of Indian Summer Monsoon is an important indicator to distinguish the monsoon season. Here the intensity of Indian Summer Monsoon (ISM) was indicated by the ISM Index, which are defined by the negative outgoing longwave radiation anomalies (with respect to the climatological annual cycle) averaged over the Bay of Bengal–India region (10°–25°N, 70°–100°E) (Wang and Fan, 1999). As shown in Fig. S1, the measurement periods (green boxes) were in the pre-monsoon season (March-May) and monsoon season (June-September), respectively. And the IMS index during the two measurements periods were equivalent to those of the whole pre-monsoon season (average: -19.5 vs -20.7 W m-2) and monsoon season (average: 27.0 vs 26.3 W m-2), respectively.

Secondary, the characteristics of meteorology and atmospheric pollutants in the two measurements periods was generally in agreement with the previous long-term studies at Nam Co station and other sites in the Tibetan Plateau (TP) (Yin et al., 2017; Cong et al., 2015; Bonasoni et al., 2010; Xu et al., 2018). That is, the characteristics of meteorology (temperature, WS and RH) and atmospheric pollutants (PM, BC and ozone) in the two measurements periods were matched with those in the whole pre-monsoon and monsoon season at Nam Co station and other sites in the TP.

Therefore, the two observation periods are representative in the seasonal characteristics in pre-monsoon season and monsoon season, respectively.

[Figure]

**Figure S1.** The Indian Summer Monsoon (ISM) Index in 2019. The measurements periods are marked by the green boxes."

*3. The authors grouped their data into "NPF-pre days", "NPF-monsoon days" and "non-event days" in*

***Fig. 3- Fig. 7 and related discussions. I may suggest separating "non-event days" into pre-monsoon and monsoon non-event days.***

Thanks for the comment. There were no typical "non-event days" in monsoon season according to the classification of NPF events. The two days which were not "NPF days" in monsoon season were "undefined days", as in which the increase of $PN_{3-10}$ was observed without the particles growing to the larger size (24 June, 2019), or we can see the later phase of a mode growing in the Aitken mode size range (18 June, 2019) (Dal Maso et al., 2005). The particle number size distributions in 18 June and 24 June, 2019 are present in Fig. R5b and c, and a typical NPF event in 23 June, 2019 is shown in Fig. R5a.

Dal Maso, M., Kulmala, M., Riipinen, I., and Wagner, R.: Formation and growth of fresh atmospheric aerosols: Eight years of aerosol size distribution data from SMEAR II, Hyytiälä, Finland, Boreal Environment Research, 10, 323-336, 2005.

[Figure]

**Figure R5.** The particle number size distribution in (a) 23 June, 2019 (NPF day), (b) 18 June, 2019 (undefined day), and (c) 24 June, 2019 (undefined day).

At the same time, we also analyzed the atmospheric characteristics in "undefined days" in monsoon season (Fig. R6). And we found that the atmospheric characteristics including CS, $J(O^1D)$, temperature, RH, wind speed, water content in "undefined days" in monsoon season were comparable with those in

"NPF days" in monsoon season. Considering that the purpose is to find the difference between "NPF days" and "non-event days", we did not add the two "undefined days" in the manuscript.

[Figure]

**Figure R6.** The Diurnal variations of (a) condensation sink (CS), (b) $JO^1D$, (c) temperature, (d) RH, and (e) $H_2O$ in NPF-pre days, NPF-monsoon days, non-event days and undefined days in monsoon season (undefined-monsoon days).

---

## Author Response (AR2)

**Reply to comments**

Journal: Atmospheric Chemistry and Physics

Manuscript Number: acp-2022-440

Title: "High frequency of new particle formation events driven by summer monsoon in the central Tibetan Plateau, China"

Author(s): Lizi Tang, Min Hu, Dongjie Shang, Xin Fang, Jianjiong Mao, Wanyun Xu, Jiacheng Zhou, Weixiong Zhao, Yaru Wang, Chong Zhang, Yingjie Zhang, Jianlin Hu, Limin Zeng, Chunxiang Ye, Song Guo, Zhijun Wu

**I. Reply to Reviewer 1**

**Reply to Reviewer 1's overall comments:**

*This revised version of the manuscript is greatly improved compared with the original one. There are, however, some minor issues that need to be considered before I can recommend acceptance of the paper for publication.*

We appreciate the comments from the reviewer on this manuscript. We have answered them in the following paragraphs (the texts italicized are the comments, the texts indented are the responses, and the texts in blue are revised parts in new manuscript). In addition, all changes made are marked in the revised manuscript. Thanks for the reviewer's affirmation on our work.

**Reply to Reviewer 1's important comments (5):**

*1. Concerning the calculation of CCN concentrations, I suggest reformulating as "…was assumed to be equal to 0.12 throughout the …" (lines 356-357). Furthermore, the statement on lines 360-361 is scientifically incorrect: a paper published in 2018 cannot say anything on measurements conducted in 2019 (the data of this paper). Please reformulate.*

Thanks for the comment. We have reformulated the expressions in the revised manuscript as follows:

"The hygroscopicity parameter κ was assumed to be equal to a constant value of 0.12 throughout the measurement period according to the previous measurement at Mt. Yulong in the TP."

"There could be uncertainties in the values of κ due to the variation of chemical components, but they had little impact on $D_c$ thus the final result of CCN concentration."

*2. Concerning the tense, one should write this suggests/indicates, not suggested/indicated (several places*

*in the text), … are shown … (line 195), … is comparable with … (line 206), … and represents (line 207), …*

*is comparable with (lines 230, 237 and 240), … results are most … (line 349), Fig. 9 shows … (line 382)*

Thanks for the comment. We have made correction in the revised manuscript.

*3. When discussing the different seasons, one should add "the" (e.g. in the monsoon seasons etc.). Also, I would write "the Nam Co station".*

Thanks for the comment. We have made correction in the revised manuscript.

*4. To make the text more readable, the numerical values of CS could be written in the form 0.02 etc rather than using an exponential form (several places in the text).*

Thanks for the comment. We have made correction in the revised manuscript.

*5. Lines 23 and 413: I would recommend reformulating: …extreme/evident seasonal differences, with 15% and… Also, decide whether this difference is extreme or evident, as these two are quite different characteristics.*

Thanks for the comment. We have reformulated the expressions in the revised manuscript as follows:

"The frequencies of NPF events exhibited **evident seasonal differences** with 15% in the pre-monsoon season and 80% in the monsoon season."

"The most important finding of this study was that there were **evident seasonal differences** in the frequencies of NPF events at the Nam Co station with 15% in the pre-monsoon season and 80% in the monsoon season."

*Line 60: … but not to biogenic…*

Thanks for the comment. We have made correction in the revised manuscript as follows:

"At Mt. Yulong on the southeastern TP, the NPF frequency was only 14% during the pre-monsoon season and the occurrence of NPF events was related to an elevated boundary layer or transported biomass burning pollutants from southern Asia, **but not to biogenic** condensable vapours (Shang et al., 2018; Du et al., 2015)."

*Line 63: … may be associated with …*

Thanks for the comment. We have made correction in the revised manuscript as follows:

"These results indicated that the frequency and mechanism of NPF **may be associated with** air mass origins and monsoon shift in the southern, southeastern and northeastern TP."

*Line 54: A significant seasonal.*

Thanks for the comment. We have made correction in the revised manuscript as follows:

"**A significant seasonal** variation of NPF frequency was observed in the TP."

*Lines 83-84: The measurements were conducted …*

Thanks for the comment. We have made correction in the revised manuscript as follows:

"**The measurements were conducted** from 26 April to 22 May, 2019 and 15 June to 25 June, 2019, and can be representative of the pre-monsoon season and the summer monsoon season, respectively (Text S1) (Bonasoni et al., 2010;

Cong et al., 2015)."

*Lines 187: the temperature … values … values …*

Thanks for the comment. We have made correction in the revised manuscript as follows:

"As shown in Fig. 2 and Fig. S9, **the temperature** behavior was characterized by higher **values** in the monsoon season (10.4$\pm$4.1 ℃) and lower **values** in the pre-monsoon season (3.1±3.6 ℃) with an average value of 5.3±5.1 ℃."

*Lines 191-192: The wind speed …. The wind direction*

Thanks for the comment. We have made correction in the revised manuscript as follows:

"**The wind speed** (WS) was comparable during the two seasons, which was 4.2±2.7 m s$^{-1}$ in the pre-monsoon season and 4.5±2.7 m s$^{-1}$ in the monsoon season, respectively. **The wind direction** (WD) showed a clear divergence, with westerly and southwesterly winds prevailing in the pre-monsoon season, and southerly winds prevailing in the monsoon season (Fig. S10).

*Line 201: … similar to …*

Thanks for the comment. We have made correction in the revised manuscript as follows:

"On average $PM_{0.8}$ was 1.8±1.0 µg m$^{-3}$, which was **similar to** $PM_1$ (2 µg m$^{-3}$) measured by a high-resolution time- of-flight aerosol mass spectrometer at the Nam Co station in 2015 (Xu et al., 2018a).

*Lines 221-222: …lower than that at … comparable with that at …*

Thanks for the comment. We have made correction in the revised manuscript as follows:

"The frequency at the Nam Co station during the pre-monsoon season was **lower than that at** NCO-P (38%)

(Venzac et al., 2008) and **comparable with that** at Mt. Yulong on the southeastern TP (14%) (Shang et al., 2018) in the same season."

*Line 276: … photochemical oxidation rate … (be consistent with the text on line 284)*

Thanks for the comment. We have made correction in the revised manuscript as follows:

   "The concentration of sulfuric acid in the atmosphere is related to the degree of $SO_2$, **photochemical oxidation**

   **rate** and CS."

*Line 287: … concentrations … levels …*

Thanks for the comment. We have made correction in the revised manuscript as follows:

   "With speculated comparable/lower $SO_2$ **concentrations** and similar CS and $J (O^1D)$ **levels**, the abundance of

   gaseous sulfuric acid in NPF days would be approaching, or little lower than that in non-event days."

*Line 312: should it rather be: … most probable reasons …*

Thanks for the comment. We have made correction in the revised manuscript as follows:

   "While the concentrations of organic precursors could be the **most probable reasons** for the occurrence of NPF

   events, the external factors driving the difference in VOC levels between the NPF and non-event days and other

   conditions that may affect the characteristics of NPF were still unknown."

*Lines 316-317: Air pollutants …site are mainly …*

Thanks for the comment. We have made correction in the revised manuscript as follows:

   "Air air pollutants at the observation site **are mainly** brought by air mass transmission."

*Line 319: … on non-event days …*

Thanks for the comment. We have made correction in the revised manuscript as follows:

   "It can be found that the dominant air masses **on non-event days** were from the west (almost 100%) and passed by

   western Nepal, northwest India and Pakistan."

*Line 332: … thus triggering …*

Thanks for the comment. We have made correction in the revised manuscript as follows:

   "The summer monsoon can bring the higher organic concentrations in the monsoon season (NPF-monsoon days)

   compared with those in the pre-monsoon season (NPF-pre and non-event days) (Fig.4), **thus triggering** almost daily

   NPF events.

*Line 378: "in a short time" is a bit vague expression. Please be more specific. I suppose you refer to the*

*few hour or bit more after NPF.*

Thanks for the comment. We have reformulated the expressions in the revised manuscript as follows:

"In addition to the average particle number concentration in the two seasons, the important impact of NPF events is more reflected in the increased number concentration of aerosol and CCN **within a few hours after particle nucleation and growth**, that is, the aerosol and CCN production."

**II. Reply to Reviewer 2**

**Reply to Reviewer 2's overall comments:**

*First of all, I appreciate the tremendous efforts the authors have put into the revised manuscript. Obviously, this manuscript has been significantly improved.*

*In the interactive discussion, the referees have raised two major concerns: 1) the representativeness of the measurement periods to the pre-monsoon and monsoon seasons; 2) the validation of modeled SO2 and VOC concentrations. Out of them, I think the authors have well addressed the first concern; yet additional evidence is needed for the latter one. I have a few suggestions for the authors' consideration.*

We appreciate the comments from the reviewer on this manuscript. We have answered them in the following paragraphs (the texts italicized are the comments, the texts indented are the responses, and the texts in blue are revised parts in new manuscript). In addition, all changes made are marked in the revised manuscript. Thanks for the reviewer's affirmation on our work.

**Reply to Reviewer 2's comments (2):**

*1. The validation of SO2 simulation. There are SO2 data available in Tibet measured at other time, which can be used to validate their model. Note that, the good correlation of SO2 and BC found at another site does not necessarily apply to the location of this study. This is because BC is inert in the atmosphere, while SO2 is quite reactive. Assuming that they are emitted by the same source, the ratio of [SO2]/[BC] would gradually decrease along with photochemical aging, which will deteriorate the correlation.*

Thanks for the comment. The comparison between simulated and observed $SO_2$ at Mt. Yulong on the southern TP has been added to validate the model in the revised manuscript. The statistical metrics of NMB (normalized mean bias) and NME (normalized mean error) values are within the range reported in previous $SO_2$ modelling result (Mao et al., 2022). The correlation coefficient (R) between simulated and observed $SO_2$ is 0.44, which reflected that the model can fairly simulate the variation of $SO_2$ concentration in Tibet.

Mao, J., Li, L., Li, J., Sulaymon, I. D., Xiong, K., Wang, K., Zhu, J., Chen, G., Ye, F., Zhang, N., Qin, Y., Qin, M., and Hu, J.: Evaluation of Long-Term Modeling Fine Particulate Matter and Ozone in China During 2013–2019, Frontiers in Environmental Science, 10, 10.3389/fenvs.2022.872249, 2022.

"**2.3 Model simulation**

For $SO_2$, the WRF/CMAQ models have been successfully reproduced $SO_2$ in major regions in China with R of 0.25-0.79 (Mao et al., 2022). And the WRF/CMAQ models achieved good performance in simulating $SO_2$ at Mt. Yulong on the southern TP (Text S2)."

"**Text S2 Model simulation**

**Model Evaluation**

The comparison between simulated and observed $SO_2$ at Mt. Yulong on the southern TP is shown in Fig. S6, which helps to validate the model performance. As shown in Table S2, the statistical metrics of NMB (normalized mean bias) and NME (normalized mean error) values are within the range reported in previous $SO_2$ modelling result (Mao et al., 2022). The correlation coefficient (R) between simulated and observed $SO_2$ is 0.44, which reflected that the model can fairly simulate the variation of $SO_2$ concentration in Tibet.

[Figure]

Figure S6. Comparison of simulated and observed SO2 (ppb). SO2 is hourly mean concentration.

Table S2. Model performance of the air pollutants at Nam Co station

| | PM$_1$ | | | O$_3$ | | | VOC | | | SO$_2$[a] | | |
|---|---|---|---|---|---|---|---|---|---|---|---|---|
| | MFB | MFE | R | NMB | NME | R | MFB | MFE | R | NMB | NME | R |
| Statistic | 0.49 | 0.50 | 0.72 | 0.14 | 0.23 | 0.51 | -0.47 | 0.49 | 0.41 | -0.44 | 0.50 | 0.44 |
| Benchmarks | <±0.6 | <0.75 | >0.4 | <±0.15 | <0.35 | >0.5 | | | | | | |
| References | | | | | | | <±0.77 | <0.74 | | <±4.38 | <±4.38 | 0.25-0.79 |

*NMB: normalized mean bias; NME: normalized mean error; R: correlation coefficient; MFB: mean fractional bias; MFE:*

*mean fractional error. The benchmarks for PM and O$_3$ were suggested by Emery et al. (2017) and Boylan and Russell (2006),*

*respectively. The references for VOC and SO$_2$ were from Hu et al. (2017) and Mao et al. (2022), respectively.*

[a] The statistical metrics for evaluating $SO_2$ simulation at Mt. Yulong on the southern TP

*2. The validation of VOC simulation. I understand that this may be a hard task for the authors. The authors*

*mention that 99 VOCs have been measured during the pre-monsoon season, which covers both NPF days*

*and non-NPF days. The author can further look into these measured VOC, focusing on the comparison of*

*VOC concentrations in NPF and non-NPF days. This would give a good hint. Also, I agree with the referee*

*that total VOC concentration is not a good quantity, because most VOCs (and especially small VOC molecules) are just spectators of NPF. The authors should pay special attention to VOCs such as monoterpene, sesquiterpene, and heavy aromatics during further analyses.*

Thanks for the comment. The comparison of VOC concentrations in NPF and non-NPF days during the pre-monsoon season has been discussed in the manuscript. It is a pity that the monoterpene and sesquiterpene were not measured in this study. The aromatics including benzene, toluene, Styrene and trimethylbenzene were measured in this study. The total concentration of aromatics exhibited a higher level (20%) during the occurrence time of NPF events in NPF-pre days compared with that in non-event days. The aromatics have been considered to contribute substantially to new particle formation (Molteni et al., 2018). The potential NPF precursors such as toluene (Garmash et al., 2020), styrene (Yu et al., 2022) and trimethylbenzene (Molteni et al., 2018) showed higher values in NPF-pre days compared with those in non-event days It gives a good hint for the role of organics in the occurrence of NPF events at the Nam Co station.

Molteni, U., Bianchi, F., Klein, F., El Haddad, I., Frege, C., Rossi, M. J., Dommen, J., and Baltensperger, U.: Formation of highly oxygenated organic molecules from aromatic compounds, Atmos. Chem. Phys., 18, 1909-1921, 10.5194/acp-18-1909-2018, 2018.

Garmash, O., Rissanen, M. P., Pullinen, I., Schmitt, S., Kausiala, O., Tillmann, R., Zhao, D., Percival, C., Bannan, T. J., Priestley, M., Hallquist, Å. M., Kleist, E., Kiendler-Scharr, A., Hallquist, M., Berndt, T., McFiggans, G., Wildt, J., Mentel, T. F., and Ehn, M.: Multi-generation OH oxidation as a source for highly oxygenated organic molecules from aromatics, Atmos. Chem. Phys., 20, 515-537, 10.5194/acp-20-515-2020, 2020.

Yu, S., Jia, L., Xu, Y., and Pan, Y.: Formation of extremely low-volatility organic compounds from styrene ozonolysis: Implication for nucleation, Chemosphere, 305, 135459, https://doi.org/10.1016/j.chemosphere.2022.135459, 2022.

"**3.3.2 Gas precursors**

Due to instrument status, VOC measurement was only available in the pre-monsoon season. The concentration of VOC (total VOC) showed a higher value (20%) during 11:00-18:00 on NPF-pre days compared with non-event days (Fig. 3c). Aromatics, which can be used as the indicator of anthropogenic emissions, also exhibited a higher level (20%) during NPF-pre days (Fig. 3d). The aromatics have been considered to contribute substantially to new particle formation (Molteni et al., 2018). The potential NPF precursors such as toluene (Garmash et al., 2020), styrene (Yu et al., 2022) and trimethylbenzene (Molteni et al., 2018) showed higher values in NPF-pre days compared with those in non-event days (Fig. S15). This suggested that VOC such as aromatics may be the key factor in determining the occurrence of NPF events."

[Figure]

**Figure 3.** Diurnal variations of (a) condensation sink (CS), (b) $JO^1D$, the total concentration of (c) VOC and (d) aromatics in

NPF-pre days, NPF-monsoon days and non-event days. The upper and lower bars indicate the 75th and 25th percentiles, the markers are the average values.

[Figure]

**Figure S15.** Diurnal variations of concentration of (a) tolunene, (b) styrene and (c) trimethylbenzene in NPF-pre days and non-event days. The upper and lower bars indicate the 75th and 25th percentiles, the markers are the average values."